# GradPS: Resolving Futile Neurons in Parameter Sharing Network for Multi-Agent Reinforcement Learning

Haoyuan Qin [1 2 *]  Zhengzhu Liu [1 2 *]  Chenxing Lin [1 2]  Chennan Ma [1 2]  Songzhu Mei [3]  Siqi Shen [1 2]
Cheng Wang [1 2]

## Abstract

Parameter-sharing (PS) techniques have been widely adopted in cooperative Multi-Agent Reinforcement Learning (MARL). In PS, all the agents share a policy network with identical parameters, which enjoys good sample efficiency. However, PS could lead to homogeneous policies that limit MARL performance. We tackle this problem from the angle of gradient conflict among agents. We find that the existence of futile neurons whose update is canceled out by gradient conflicts among agents leads to poor learning efficiency and diversity. To address this deficiency, we propose GradPS, a gradient-based PS method. It dynamically creates multiple clones for each futile neuron. For each clone, a group of agents with low gradient-conflict shares the neuron's parameters. Our method can enjoy good sample efficiency by sharing the gradients among agents of the same clone neuron. Moreover, it can encourage diverse behaviors through independently updating an exclusive clone neuron. Through extensive experiments, we show that GradPS can learn diverse policies with promising performance. The source code for GradPS is available in https://github.com/xmu-rl-3dv/GradPS.

## 1. Introduction

Many real-world tasks can be modeled as cooperative Multi-Agent Reinforcement Learning (MARL) problems (Hernandez-Leal et al., 2019), where multiple agents must collaborate to achieve a common goal (Rashid et al.,

*Equal contribution [1]Fujian Key Laboratory of Sensing and Computing for Smart Cities, School of Informatics, Xiamen University (XMU), China [2]Key Laboratory of Multimedia Trusted Perception and Efficient Computing, XMU, China [3]School of Computer, National University of Defense Technology, China. Correspondence to: Siqi Shen <siqishen@xmu.edu.cn>.

*Proceedings of the 42nd International Conference on Machine Learning*, Vancouver, Canada. PMLR 267, 2025. Copyright 2025 by the author(s).

2020). MARL is challenging due to issues such as non-stationary (Qiu et al., 2021), partial observability (Oliehoek & Amato, 2016), scalability, credit assignment (Foerster et al., 2018; Rashid et al., 2018), low sample efficiency, etc. Researchers have proposed various methods to address these issues. Many of them adopted the Centralized Training with Decentralized Execution (CTDE) paradigm (Oliehoek et al., 2008), where agents are trained jointly with central information, but the agents execute their policies decentrally with the agent's local observation. In the CTDE paradigm, Parameter Sharing (PS) (Rashid et al., 2018) is a widely adopted technique in MARL to alleviate the scalability and low sample efficiency issues.

In PS, all agents share a neural network policy determined by the same set of parameters. PS is effective for MARL tasks where the agent behaviors are identical or similar. However, PS can lead to homogeneous policies, limiting the diversity of agent behaviors and the overall capabilities of joint policies.

Existing studies on PS can be classified into four categories: (1) full PS approaches where all agents share the same parameters, the identity of an agent is used as input to the policy to increase policy diversity (Rashid et al., 2018; Qiu et al., 2021; Wang et al., 2020), (2) group-based PS where agents are group (either dynamically or statically) into multiple groups based on the behaviors (e.g., trajectories), each group shares the same set of parameters (Christianos et al., 2021; Li et al., 2024b). (3) partial PS where the neural network of the policy is divided into a shared part and an individual part (Li et al., 2021), (4) mask-based PS where the agent policy is determined by the combination of a shared policy network and an agent-specific mask that can be static (Kim & Sung, 2023) or dynamic (Li et al., 2024d). Our work is a neuron-based PS approach that addresses the problems of PS from the angle of neuron gradient conflict among agents.

Neuron gradient refers to the gradient of a loss with respect to a neuron. The gradient of the neuron represents the change in the knowledge learned by the agent. In a PS network, agents could lead different knowledge which lead to gradient conflict. The neuron gradient conflict indicates

gradient directions for a neuron among some agents are different. Regarding learning efficiency, gradient conflict is harmful, as it cancels out the learning progress of agents. Regarding policy diversity, gradient conflict could be beneficial, suggesting that agents have learned different skills and knowledge.

In this work, we study the gradient conflict in PS networks. We find the existence of futile neurons whose updates are canceling out significantly by gradient conflict. We find that these futile neurons can affect the expressiveness of the MARL policy network.

We propose a method, Gradient-based Parameter Sharing (GradPS), to resolve futile neurons in the PS network. GradPS records the gradient conflicts between agents for the last $T$ interval during training. For each futile neuron, GradPS dynamically creates $K$ clones for the neuron. For each clone, a group of agents with low gradient conflict shares the parameters of the clone. Its parameters are updated according to the gradient of the agents in the group. A group can consist of multiple agents, and their gradients (knowledge) with respect to the neuron can be shared among the group. A group can consist only of a few agents, which helps the agents to develop their specialized skills. As agent policies evolve, cloned neurons assigned to groups may once again become futile, while previously conflicting agents may no longer exhibit conflicts. To adapt to these changes, we re-evaluate inter-agent conflicts and restore a subset of cloned neurons with minimal conflicts to their original state, enabling a dynamic regrouping process.

Through extensive experiments on multiple benchmarks, we show that GradPS performs better than state-of-the-art PS methods. GradPS can learn diverse policies through utilizing the gradient conflict information.

## 2. Background

### 2.1. Dec-POMDPs

We consider Decentralized Partially Observable Markov Decision Processes (Dec-POMDPs) (Oliehoek & Amato, 2016) in modeling cooperative multi-agent reinforcement learning (MARL) scenarios. A Dec-POMDP can be formally described by the tuple $G = \langle \mathcal{S}, \{\mathcal{U}_i\}_{i=1}^{N}, P, r, \{\mathcal{O}_i\}_{i=1}^{N}, \{\sigma_i\}_{i=1}^{N}, N, \gamma \rangle$, where $\mathcal{N}$ represents the set of agents, $\mathcal{S}$ is a finite set of states, and $\mathcal{U}_i$ is the set of actions available to agent $i$. At time step $t$, each agent $i$ chooses an action $u_i^t \in \mathcal{U}_i$, forming a joint action $\boldsymbol{u}^t \in \mathcal{U}^N = \mathcal{U}_1 \times \ldots \times \mathcal{U}_N$. This leads to a transition to a new state $s^{t+1} \sim P(\cdot|s^t, \boldsymbol{u}^t)$ and a joint reward $r^t$. In consideration of partial observability, each agent can only access an individual observation $o_i^t \in O_i$, which is drawn from $o_i^t \sim \sigma^i(\cdot|s^t)$. $\gamma$ denotes the discounting factor. Each agent acts base on individual policy

$\pi_i(u_i|\tau_i)$, $\tau_i = (O_i \times U_i)^*$ represents agent's local action-observation history. the global action-observation history is denoted as $\tau \in \mathcal{T}^N := \tau_1 \times \ldots \times \tau_N$, on which it conditions the joint policy $\pi = <\pi_1, \ldots, \pi_N>$. The joint policy $\pi$ has a joint action-value function: $Q^\pi(s_t, \boldsymbol{u}_t) = \mathbb{E}_{s_{t+1:\infty}, \boldsymbol{u}_{t+1:\infty}}[R_t \mid s_t, \boldsymbol{u}_t]$, where $R_t = \sum_{i=0}^{\infty} \gamma^i r_{t+i}$ is the discounted return.

### 2.2. Gradient Conflict

Gradient conflict is a common phenomenon in multi-task learning (MTL), where multiple different but related tasks are jointly trained by sharing a model (Caruana, 1997). One implementation of MTL is to jointly train a network for all tasks, with the goal of discovering and leveraging relationships between tasks.

Learning multiple tasks may result in worse overall performance and data efficiency. One reason for this is gradient conflict where the gradients of multiple tasks diverge, such that the update for one task negatively affects the other. Researchers (Yu et al., 2020) consider the gradient conflict for a vector, which is defined as follows.

**Definition 1** (Vector Conflicting Gradients (Yu et al., 2020))**.** *For a vector, the gradients $g_i$ and $g_j$ $(i \neq j)$ for tasks $i$ and $j$ are said to be conflicting with each other if $\cos \phi_{ij} < 0$, where $\phi_{ij}$ is the angle between $g_i$ and $g_j$.*

Conflicting gradients pose a challenge to optimizing the multi-task objective, as different gradients of individual tasks may impede the learning of knowledge.

Here, vector gradient conflict refers to the conflict between gradient vectors composed of neuron gradients across different tasks in a neural network. In contrast, the neuron gradient conflict proposed in this paper specifically pertains to conflicts in gradient values for a particular neuron among different agents within a PS network.

## 3. Related Work

### 3.1. Parameter Sharing (PS)

**Multi-Agent PS** Parameter sharing has been widely used in MARL algorithms due to its simplicity and high sample efficiency. However, it could lead to homogeneous agent behaviors. ROMA (Wang et al., 2020) adopts a full PS approach, where agent behaviors are encoded into roles represented by a hidden vector. Agent policies are sampled from the hidden vector. SePS (Christianos et al., 2021) adopts a group-based PS approach by creates multiple sets of parameters each shared by a group of agents. Agents are clustered into groups based on their behaviors (trajectories) in the beginning of training. SNP (Kim & Sung, 2023) employs a partial PS approach. It creates a big neural network, and then each agent builds its policy based on part of the

network through network pruning. AdaPS (Li et al., 2024b) combines SNP and SePS by proposing a cluster-based partial PS approach. Kaleidoscope (Li et al., 2024d) adopts a mask-based PS approach, an agent policy is determined on both a full PS network and a mask, which is learned dynamically for each agent.

**Multi-Task PS** Parameter sharing is also widely used in multi-task learning (MTL), which learns a single model for multiple different tasks. By PS among different tasks, MTL methods can learn more efficiently with an overall smaller model size compared to learning with separate models. PCGrad (Yu et al., 2020) found that parameter sharing of different tasks in MTL produces gradient conflicts and hurt performance. To avoid gradient conflicts, CAGrad (Yu et al., 2020) seeks an update vector that maximizes the worst local improvement of any target within the average gradient neighborhood, optimizing the minimum reduction rate of any specific task loss. Gradient Vaccine (Wang et al., 2021c) uses task relevance to set gradient similarity goals and adaptively aligns task gradients to achieve these goals. GradNorm (Chen et al., 2018) balances training in deep multi-task models by dynamically adjusting the gradient magnitude among tasks. Recon (Guangyuan et al., 2023) selects the layers with high conflict scores and turns them into task-specific layers.

### 3.2. Neuron Learning Efficiency

Redo (Sokar et al., 2023) identifies the existence of dormant neurons, whose activation score is low during the training procedure. The dormant neurons lead to poor learning efficiency. Reset (Igl et al., 2021; Nikishin et al., 2022) improves the plasticity of neural networks through periodic resets of the weights of neurons in the last layer of a neural network. (Dohare et al., 2024) proposes continue backpropagation algorithm which periodically re-initiates some less-used neurons. Reborn (Qin et al., 2024) perturbs parameters of dormant neurons in MARL mixing network.

Our work analyzes the learning efficiency of MARL neurons based on gradient among different agents. Moreover, we identify the existence of futile neurons, a special type of neurons whose update are canceled mostly by gradient conflicts.

## 4. The Futile Neuron Phenomenon in MARL

In this section, we study the gradient conflict in the Parameter Sharing (PS) network for MARL. We find the existence of gradient conflicts in PS, identify futile neurons whose updates are affected significantly by the gradient neurons, and find that futile neurons hurt MARL network expressiveness.

We study the most popular PS (Sunehag et al., 2018; Wang et al., 2021a; Rashid et al., 2018; Shen et al., 2022), where

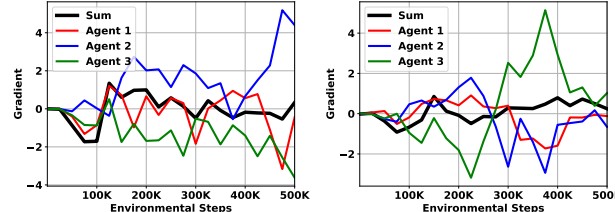

Figure 1: Gradients of a neuron for different agents: VDN (Left) and QMIX (Right) in the SMAC 3m environment.

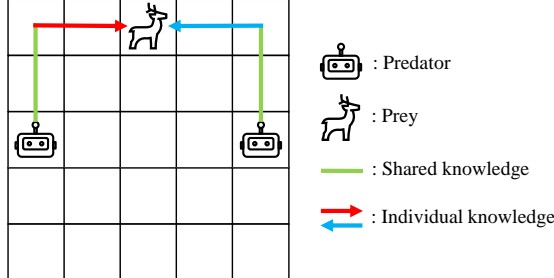

Figure 2: Two agents collaborate to capture a prey in a predator-prey environment.

the parameters of the agent policy are shared among all agents. The observation of each agent is concatenated with its agent identity to increase policy diversity. The neural network of the PS agent network consists of two fully-connected layers and a GRU layer. We analyze the neurons in the fully connected layers of the agent network.

### 4.1. Neuron Gradient Conflict Exists in MARL

In this section, we study the gradient of the temporal difference loss with respect to neurons for different agents with the VDN and QMIX methods.

**Definition 2** (Neuron Gradient). *During a target network update period, the accumulate sum of the gradients generated by the observation data of agent $i$ on neuron $n$ after backpropagation through loss is the neuron gradient $\phi_i$.*

In Figure 1, the gradient of two neurons for the SMAC 3m environment with the VDN and the QMIX method are plotted in the left and the right part of the graph, respectively. Different agents could have different gradients, which could be negative or positive. The aggregated gradient for the neuron (depicted as the sum) could become zero due to gradient conflict. For Figure 1, the RMSProp optimizer is used, we show in Appendix Figure 1 that gradient conflicts exist for the SGD and Adam optimizer too.

Gradient conflicts are common in multi-agent reinforcement learning. Even if two predators cooperate to hunt the same target, the learned knowledge may still conflict due to the

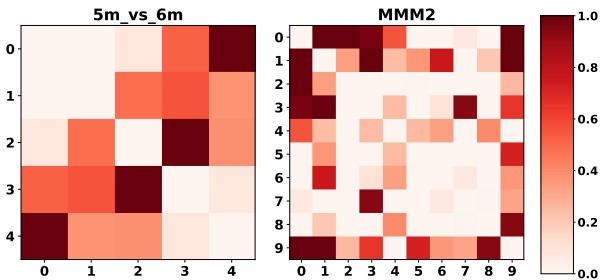

Figure 3: Average Neuron Gradient Conflict of neurons for each agent: 5m_vs_6m (Left) and MMM2 (Right).

different initial positions in the environment. As shown in Figure 2, the agents need to learn shared knowledge and individual knowledge. Differences in individual knowledge may lead to conflicts in certain neurons. The conflicting neuron gradients cancel each other out, which slows down the learning process of neurons. We define Neuron Gradient Conflict as follows.

**Definition 3** (Neuron Gradient Conflict (NGC)). *During a target network update interval $T$, for neuron $n$, the accumulated gradient of agent $i$ at the activation layer is $\phi_i^T$, and the gradient of agent $j$ is $\phi_j^T$. The gradient conflict between agents $i$ and $j$ on this neuron within the $T$ interval is defined as:*

$$C_{ij}^T = \sum_T (|\phi_i^T| + |\phi_j^T| - |\phi_i^T + \phi_j^T|) \tag{1}$$

We study the NGC $C_{ij}^T$ of all the neurons in the first FC layer for interval $T$ with QMIX. Figure 3 (left) shows the average value of the NGC for the 5m_vs_6m environment. The value of cell $(i, j)$ of the table represents the average NGC among agent $i$ and $j$. The darker the color of the cell in the table, the greater the conflict between the agents. It can be seen that there are gradient conflicts in such an environment where agents belong to the same agent type. Moreover, we find that the aggregate NGC behaviors among different agents vary. The conflict among agent 0 and 1 is low as their behavior are similar. The average NGC between agent 0 and agent 4 is high as the learned behaviors of them are different.

Figure 3 (right) shows the average NGC for the SMAC MMM2 environment, where agents have heterogeneous agent types. Agents 0 and 1 belong to marauder agents, agents 2 to 8 are marine agents, and agent 9 is a medivac agents. The agent policies should be diverse to finish such task. It can be seen that the mean NGC among all the marine agents (agents 2 to 8) are small compared to the conflicts among other agent types. Agent 9 has a large gradient conflict with most of the agents, as it is a healing agent rather than a combat agent (e.g., marine). *The NGC among agents can be used as a measure to distinguish the role/task*

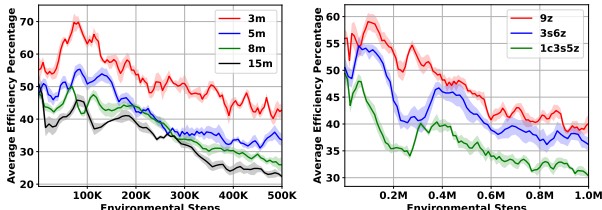

Figure 4: Neuron Gradient Efficiency in homogeneous (Left) and heterogeneous (Right) SMAC environments.

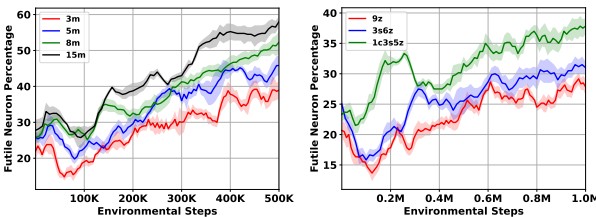

Figure 5: Percentage of Futile Neurons in homogeneous (Left) and heterogeneous (Right) SMAC environments.

*of agents during MARL task training.*

### 4.2. Futile Neurons in MARL

We study the learning efficiency of neurons in terms of the NGC among agents. It is defined as follows.

**Definition 4** (Neuron Gradient Efficiency (NGE)). *During the last $T$ periods of the target network update interval, for neuron $n$, the accumulated gradient for agent $i$ of the update interval $T$ is $\phi_i^T$. The Neuron Efficiency for neuron $n$ is defined as follows:*

$$e_n = \frac{1}{T} \sum_{t=1}^{T} (|\sum_i \phi_i^T| / \sum_i |\phi_i^T|) \tag{2}$$

NGE is a value among 0 and 1. To evaluate NGE in MARL, we conduct experiments in homogeneous SMAC environments with only one agent type. As is depicted in Figure 4 (left), *with the increase of training time, the neuron efficiency decrease*. This finding suggests that with the increase of the training time, the gradient conflicts among different agents increase, which may lead to inefficient learning. In Figure 4 (left), different environments with an increasing number of agents are considered. There are 3 and 5 agents in the 3m and the 5m environments, respectively. We find that *with increasing number of agents, the neuron efficiency decreases*. This is due to the fact that more agents could lead to more gradient conflicts.

In Figure 4 (Right), three environments with the same number (i.e., 9) of agents are shown. There are 1, 2, and 3 types of agents in the 9z, 3s6z, and 1c3s5z environments, respectively. We find that with the increase of agent type, the NGE

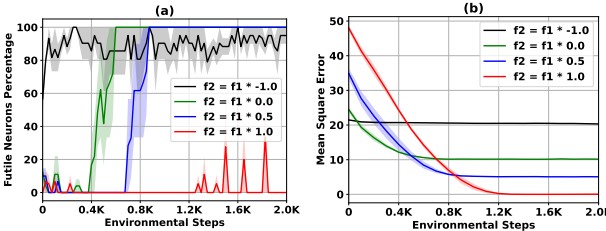

Figure 6: (a) Percentage of futile neurons, (b) MSE of reconstructed Q value

decrease. For some neurons, their NGE can be very low. Such neurons are refer to Futile Neurons defined as follows.

**Definition 5** (Futile Neuron). *A neuron $n$ is a futile neuron if it satisfies the following condition.*

$$e_n < \alpha \tag{3}$$

where $0 < \alpha < 1$ (i.e., $\alpha = 0.2$). To ensure fairly display of futile neurons across different scenarios, we fixed $\alpha$ to a constant threshold in experiments. For practical implementation, more flexible methods such as median absolute deviation (MAD) can be employed to dynamically identify futile neurons.

Figure 5 depicts the percentage of Futile Neurons for different SMAC environments. It shows that the percentage increases with time. And such percentage is non-neglectable. Figure 5 (left) shows that with an increasing agent count, the futile neuron percentage increases. Such a percentage increases with the number of agent types in Figure 5 (right).

### 4.3. Futile Neurons Could Hurt Network Expressiveness

The expressiveness of value factorization functions (Rashid et al., 2018; Son et al., 2019) is shown to significantly impact MARL performance. In this section, we conduct experiments to study whether futile neurons hurt agent network expressiveness. We conduct experiments on multiple simple one-step matrix games. In these games, there are only two agents with only one time step, all agents shared a team reward without receiving separated rewards.

We build the payoff matrix as a sum of two utility functions, and use VDN as the value factorization function, which can model the summation relationship. The payoff matrix is constructed as $Q(u_1, u_2) = f_1(u_1) + f_2(u_2)$, where $u_1$ and $u_2$ are the actions taken by agent 1 and 2, respectively, and $f_1$ and $f_2$ are two utility functions. $f_1(A) = 0$, $f_1(B) = 1$, $f_1(C) = 2$, $f_1(D) = 3$. $f_2 = w \cdot f_1$, where $w \in \mathbb{R}$. Each matrix games differ by the value $w$. Please refer to all matrices in Appendix D.2.3.

Figure 6 (left) shows the percentage of futile neurons during

the training process. When $f_1 = f_2$ ($w = 1$), there is basically no futile neuron, while in other scenarios, futile neurons gradually appear. The larger the discrepancy $w$ from 1, the earlier the appearance of futile neurons. Moreover, we find that *the time of the appearance of futile neurons correlates with the time that the MSE of the Q values is close to convergence*, as shown in Figure 6 (right). For example, at step around $0.65k$, the futile neurons for the curve $f_2 = f_1 \times 0.5$ suddenly increase. At the same time, the MSE for $f_2 = f_1 \times 0.5$ converges, and the loss does not drop thereafter.

From Figure 6, we find that the appearance of futile neurons prevents the further convergence of the MSE loss for the matrix game. The gradient conflicts generated by knowledge differences will lead to the generation of futile neurons. It is possible that the gradient conflicts for these futile neurons cancel out most of the learning process, which hurts the expressiveness of the MARL networks.

## 5. Method

In this section, we propose the GradPS method to address the futile neuron phenomenon in MARL. The overview of the GradPS algorithm is described in Algorithm 1, with a schematic diagram shown in Figure 7. GradPS groups agents into $K$ distinct groups for each futile neuron, and the parameters of group are updated independently to reduce gradient conflicts. Thus, the gradient conflict for futile neurons is alleviated. We describe the key operation of the algorithm in the following sections.

### 5.1. Conflict-Based Grouping

The appearance of futile neurons indicates significant conflicts in the agents during recent periods. Gradient conflicts Gradient conflicts correlate with diverse observations and actions of individual agents, which gradient descent methods struggle to resolve effectively. The gradients of agents with larger conflicts are more likely to cancel each other out, so we group agents with smaller conflicts based on the conflict matrix.

For each target update interval, the neuron gradient efficiency of each neurons is recorded, and futile neurons are identified. To reduce gradient conflicts among agents, we cluster agents into $K$ groups based on their gradient conflict values accumulated over $T$ periods, aiming to minimize the total gradient conflict within each group. The clustering is performed based on the average gradient conflict among agents.

Specifically, we convert the conflict matrix into an affinity matrix and use the spectral clustering algorithm for grouping the agents. Please refer to Appendix C.2 and C.3 for details about the recording of gradients and the clustering.

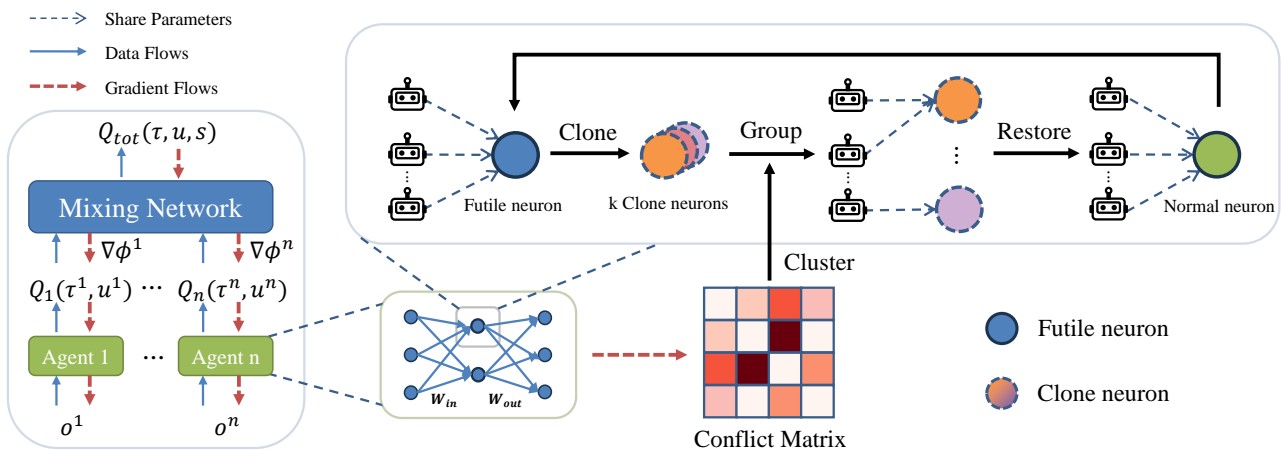

Figure 7: The schematic plot of GradPS. Each futile neurons is duplicated into $K$ clones, each is shared by distinct group of agents. If the difference among clones are small, they will be restored into a normal neuron.

---

**Algorithm 1** GradPS Algorithm

---

1: Get the accumulated gradients for all the neurons.
2: Identify the Futile Neurons
3: **for** Each futile neuron $n$ **do**
4:     Grouping agents into $K$ groups according to gradient conflicts (Sec. 5.1)
5:     Makes $K$ clones of the futile neuron $n$, and assign each clone to a distinct group of agents.(Sec. 5.2)
6: **end for**
7: **if** Restore operation required **then**
8:     Calculate the variance of each neuron
9:     Restoring the clones to a normal neuron (Sec. 5.3)
10: **end if**

---

### 5.2. Group-Based Neuron Clone

After the grouping operation, for a futile neuron $n$ in linear layer, its input weights $W_{in}$ and bias $b_{in}$ are cloned $K$ time. Each clone $j$ is shared among the group $j$ of agents, and the update of clone $j$ is independent of other clones. Each agent belongs to a distinct group.

The output of clone $j$ for the futile neuron $n$ is defined as $W_{in}^j x + b_{in}^j$. Each clone does not individually connect to the neurons in the next layer, their outputs are aggregated into $F(\sum_{j=1}^K W_{in}^j x + b_{in}^j)$ as the input for the next layer, where $F$ is the activation function.

During training, the input weights $W_{in}^j$ for the clone neuron $j$ are updated independently from other clones. As the grouping operation has made the gradient conflict in each group small, the update of $W_{in}^j$ for the clone neuron $j$ enjoys a better learning efficiency than without sharing. Moreover, agents can learn diverse policies without gradient conflict within a clone neuron. Although the efficiency and diversity improvement comes at the cost of more neurons, we will

show in the experiment section that the cost is low.

### 5.3. Group Parameter Sharing Restoring

As agent policies evolve, cloned neurons assigned to groups may once again become futile, while previously conflicting agents may no longer exhibit conflicts. To adapt to these changes, we re-evaluate inter-agent conflicts and restore a subset of cloned neurons with minimal conflicts to their original state, enabling a dynamic regrouping process.

In each restoration period, with a probability of $\rho$ times the percentage of futile neurons, we select the group of clones, whose parameter variance are smallest, for restoration. The restoration process changes the weights of each clone to the same values.

It is possible to simply set the weights of the clones as the average weights of all the groups. However, such an operation could lead to forgetting learned knowledge among different groups. To this end, we add a regularization loss for parameter restoration and use a separate SGD optimizer for gradual parameter restoration. The regularization loss is defined as follows.

$$L_{reg} = \sum_{j=1}^K \left( |W_{in}^j - \overline{W_{in}}| + |b_{in}^j - \overline{b_{in}}| \right) \quad (4)$$

where $\overline{W_{in}}$ and $\overline{b_{in}}$ are average values among groups. After $M$ updates by the SGD optimizer, when the group's parameters converge, we merge the clones and restore them into a normal neuron that is shared among all agents.

## 6. Empirical Evaluations

We show that GradPS performs better than the other parameters sharing (PS) methods for the SMAC and the Predator-Prey benchmarks. GradPS can improve the performance

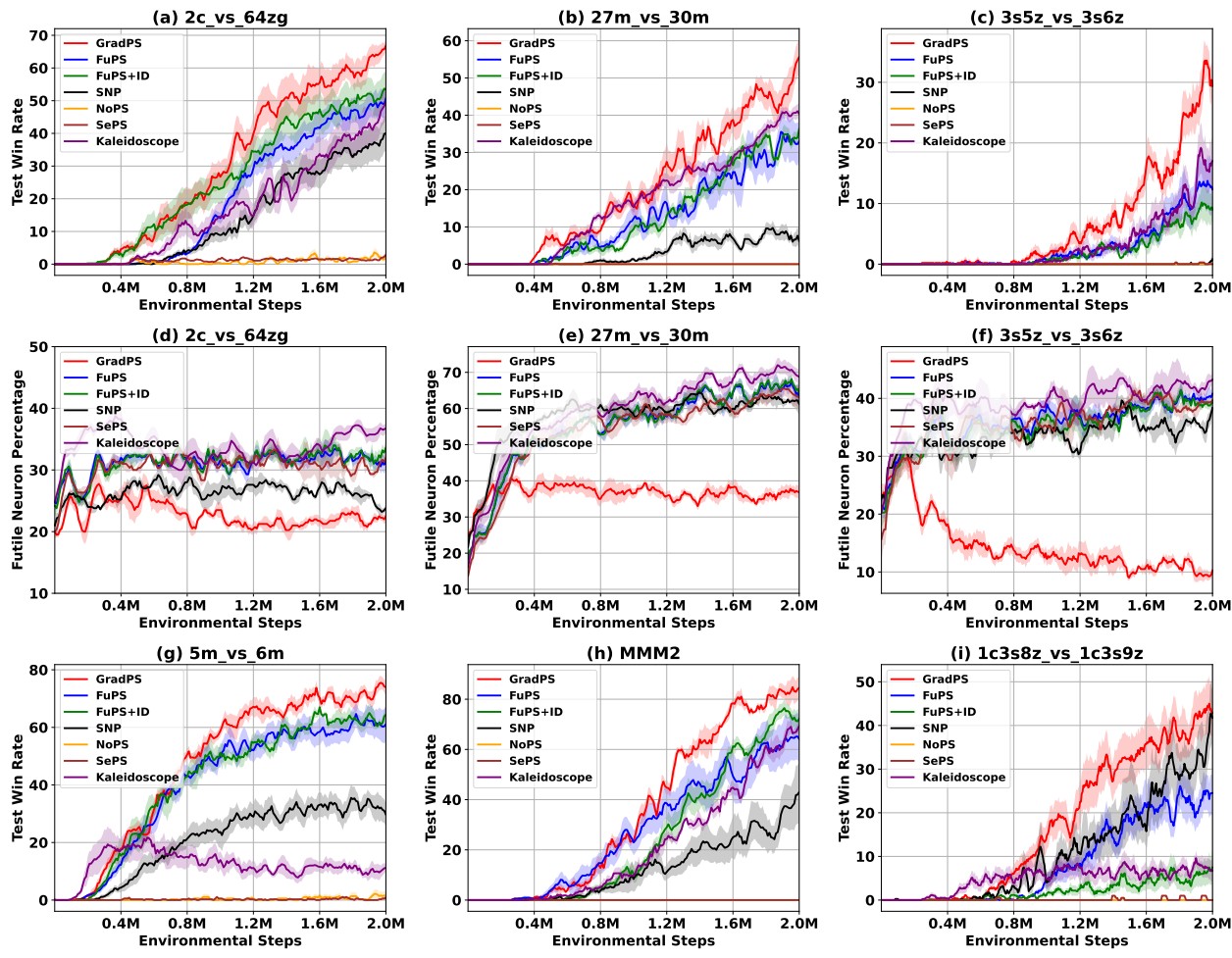

Figure 8: GradPS performs better than other PS methods: (a-c) The test win rate and the futile neuron percentage (d-f) of the 2c_vs_64zg, 27m_vs_30m, 3s5z_vs_3s6z environment. (g-i) The test win rate of the 5m_vs_6m, MMM2, and 1c3s8z_vs_1c3s9z environments.

of MARL algorithms by reducing the percentage of futile neurons in PS networks. GradPS is able to learn diverse policies thanks to identifying hidden properties of agents based on gradient conflicts, and GradPS works for multiple MARL agent networks. Please refer to Appendix D.3 for a detailed experimental setup and more experimental results.

## 6.1. Environmental Setup

**Environments. The StarCraft Multi-Agent Challenge (SMAC) (Samvelyan et al., 2019)** is a popular benchmark used extensively in MARL, where multiple ally units controlled by MARL algorithms aim to defeat enemy units controlled by the game's built-in AI. **Predator-Prey** simulates a grid world where multiple predators collaborate to capture preys, which consists of stag and hare. Capturing a stag can lead to a higher reward than capturing a hare, but it requires close collaboration between two predators. In this

experiment, some agents hold hidden property that they will receive the same reward for capturing a hare as capturing a stag. Such hidden property cannot be observed by any agents. We evaluate three environments of Predator-Prey: small, medium, and large. Each of them consists of different numbers of agents with varying grid sizes.

**Baselines and training.** We compare GradPS with various parameter sharing methods including (1) Full Parameter Sharing (FuPS), (2) Full Parameter Sharing with index (FuPS+id), (3) Selective Parameter Sharing (SePS) (Christianos et al., 2021), (4) Structured Network Pruning with parameter Sharing (SNP) (Kim & Sung, 2023), (5) Kaleidoscope (Li et al., 2024d), (6) No Parameter Sharing (NoPS). We evaluate the performance of GradPS and all these methods with the QMIX (Rashid et al., 2018) value factorization function. Detailed implementations and parameter configurations are available in Appendix D.1.

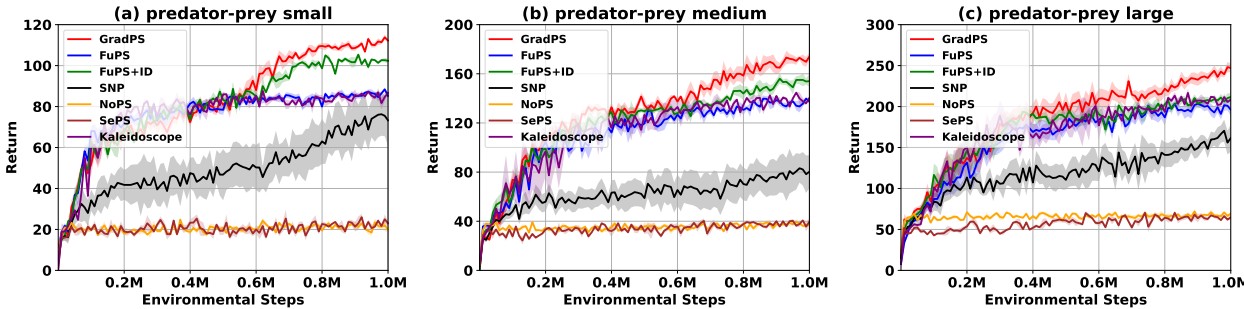

Figure 9: GradPS performs better than other PS methods on Predator-Prey: (a) Small, (b) Medium, (c) Large

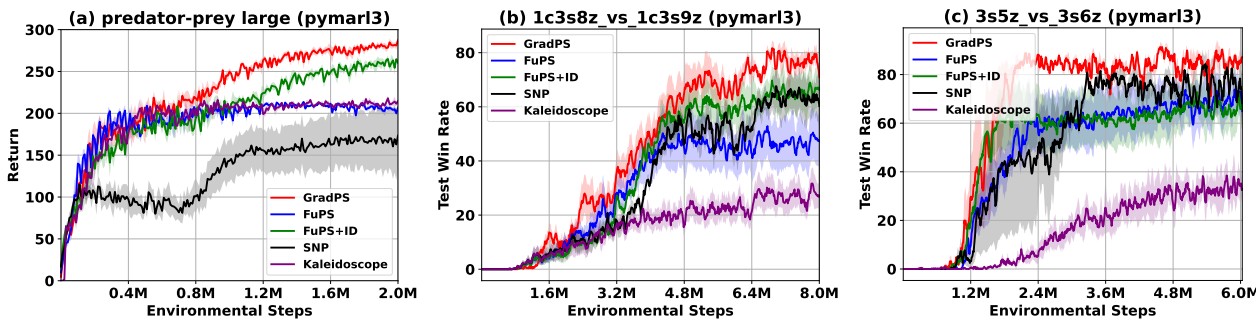

Figure 10: Performance of GradPS based on PyMARL3 with more environmental steps. (a) Predator-Prey Large, (b) 1c3s8z_vs_1c3s9z, (c) 3s5z_vs_3s6z

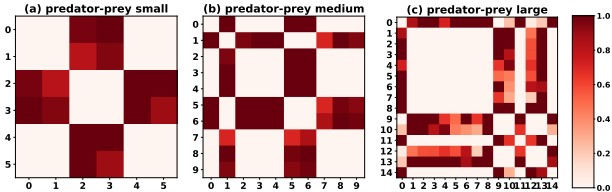

Figure 11: The average gradient conflicts among agents for the (a) small, (b) medium, and (c) large maps of the predator-prey benchmark. The darker the cell, the larger the gradient conflict. The agents with the hidden property in the small map are 2 and 3, in the medium map are 1, 5, and 6, and in the large map are 0, 9, 10, 12, and 13. Agents can be divided into two groups, with smaller conflicts within each group. GradPS successfully distinguishes agents with the hidden properties through gradient conflicts.

## 6.2. GradPS Is Superior to Other PS Methods

The experimental results for various PS methods and GradPS are shown in Figure 8. For the 2c_vs_64zg, the 27m_vs_30m, the 3s5z_vs_3s6z environments, as is shown in Figure 8 (a-c), GradPS achieves the best performance in terms of win rate thanks to its ability to reduce the futile neurons better than other PS methods, as is shown in Figure 8 (d-f). For the 5m_vs_6m, MMM2, 1c3s8z_vs_1c3s9z environments, GradPS performs the best among all the methods as well. Figure 9 depicts the results for the Predator-Prey environments. As shown in the graph, GradPS performs better than other PS methods for the Predator-Prey environments.

In addition, we implemented GradPS based on PyMARL3 and extended the environmental steps, as shown in the Figure 10. More results are presented in Appendix 7 and 8.

## 6.3. GradPS Can Encourage Diverse Policies

In real-world scenarios, there may exist unobservable features that could significantly influence agent policies. For some agents (with hidden property) in predator-prey environments, capturing a hare can lead to the same reward as capturing a stag. After carefully inspecting the learned policies, we find that agents with such hidden property learn to prefer to capture hares.

Figure 9 shows the comparison with other PS methods. Although FuPS+ID can also learn slightly different behaviors, this becomes difficult as the number of agents increases.

To analyze such a phenomenon, we depict the average gradient conflict among agents for the last $T$ update periods in Figure 11. We find that GradPS can distinguish agents with the hidden property through gradient conflicts and learn diverse policies. Agents with the hidden property have a large

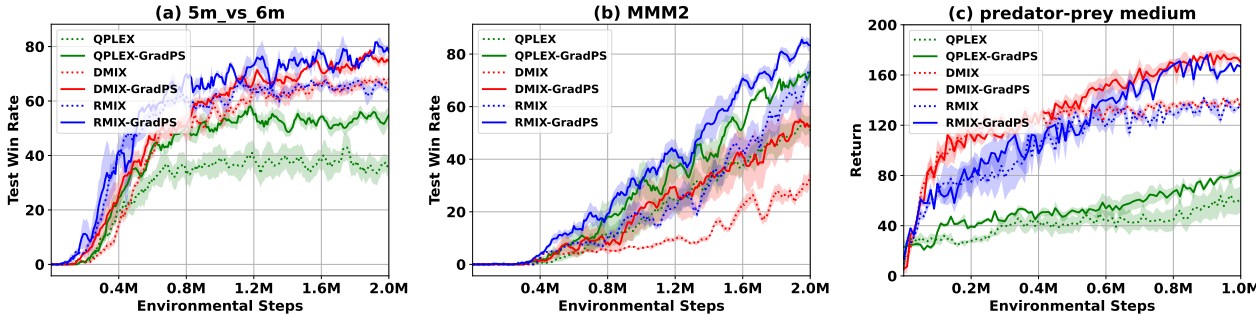

Figure 12: GradPS works for different agent networks (i.e., QPLEX, DMIX, RMIX): (a) 5m_vs_6m, (b) MMM2, (c) Predator-Prey Medium environment.

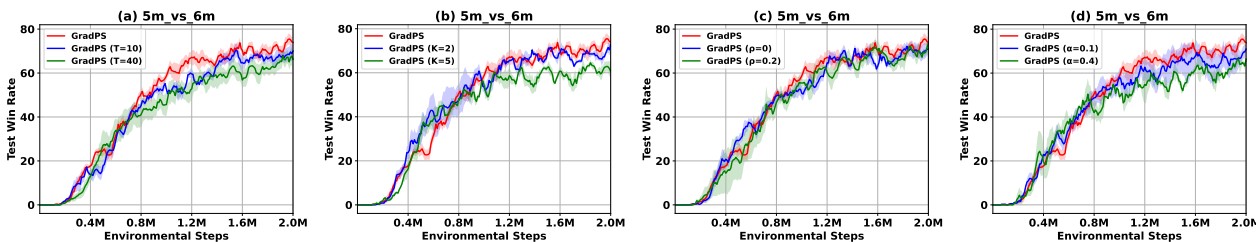

Figure 13: Ablation of different hyperparameters in GradPS. (a) the period of accumulating gradients $T$. (b) the number of groups $K$. (c) the restoration probability $\rho$. (d) the futile threshold $\alpha$.

gradient conflict with agents without the property. GradPS learns different behaviors through the use of the grouping of cloned neurons. Further, we show that GradPS leads to less futile neurons in Appendix Figure 5 (e-f).

### 6.4. GradPS Works for Multiple PS Agent Networks

In this section, we investigate the applicability of GradPS by validating GradPS's ability to enhance performance across various value factorization algorithms (QPLEX, DMIX, RMIX) in different experimental scenarios (5m_vs_6m, MMM2, Predator-Prey Medium).

The agent architectures of QPLEX, DMIX, and RMIX feature a separate value head, distributional function, and risk-sensitive function, respectively. They are different from the architecture of agents used in QMIX. According to the experimental results presented in Figures 12, GradPS can improve the performance of multiple PS agent architectures and effectively reduce gradient conflicts between agents.

### 6.5. Parameter Sensitivity Analysis

The ablation study in Figure 13 illustrates the impact of different hyperparameter settings in QMIX-GradPS, focusing on the ablation of the futile threshold $\alpha$, the probability of the restoration $\rho$, the number of group $K$, and the group interval $T$. The default configuration of QMIX-GradPS is $\alpha = 0.2$, $T = 20$, $K = 3$, and $\rho = 0.05$.

We individually modify each hyperparameter, and the experimental results indicate that appropriate hyperparameters can help better distinguish and utilize futile neurons, thereby improving learning efficiency.

The parameter $\alpha$ can be adaptively optimized rather than constrained by fixed thresholds. It is not recommended to set a large $\rho$ because gradual parameter recovery requires sufficient time to prevent rapid performance degradation. In future work, We will explore whether the aggregate $K$ of all the neurons can represent agent heterogeneity.

## 7. Conclusion

In this work, we study the gradient conflict among agents for the Parameter Sharing (PS) networks of Multi-Agent Reinforcement Learning (MARL). We identify the existence of futile neurons whose update is canceled out by gradient conflicts. We show that such neurons hurt the policy expressiveness. We propose GradPS, a simple yet effective gradient-based PS method that resolves futile neuron issues. It dynamically creates multiple independent clones for each futile neuron. Each clone is shared among a group of agents with low gradient conflicts. GradPS can learn diverse behaviors through multiple clones to avoid gradient conflict, and enjoys good sample efficiency by sharing the gradients among agents of the same clone neuron. Through extensive experiments, we show that GradPS is promising.

# Acknowledgements

This work was partially supported by the Fundamental Research Funds for the Central Universities (No. 20720230033), by Xiaomi Young Talents Program. We would like thank the anonymous reviewers for their valuable suggestions.

# Impact Statement

This paper presents work whose goal is to advance the field of Machine Learning. Our work has many potential societal consequences, none of which must be specifically highlighted here.

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

# A. Background

## A.1. Spectral Clustering

Spectral clustering (Von Luxburg, 2007; Ng et al., 2001) is a graph-based clustering method that first treats data points as nodes in a graph and computes an affinity matrix based on similarity measures such as Euclidean distance or Gaussian kernel functions. The affinity matrix is then transformed into the Laplacian matrix. The Laplacian matrix L is constructed using the graph's affinity matrix $A$ and its degree matrix $D$, as represented by the formula $L = D - A$. The Laplacian matrix captures the connectivity between nodes in the graph and reflects the structural information of the graph, playing a key role in spectral clustering. By performing spectral decomposition of the graph Laplacian, a low-dimensional embedding of the data is obtained. In this low-dimensional space, data points within the same cluster are more tightly grouped, while the distance between different clusters is larger, enabling the clustering algorithm to perform better on complex and nonlinear datasets.

Finally, a conventional clustering algorithm, such as k-means, is applied to this low-dimensional representation. Compared to traditional k-means, spectral clustering often outperforms in handling nonlinear structures and complex data, as it can better capture implicit relationships between data points.

# B. Related work

## B.1. Multi-Agent Reinforcement Learning

Multi-Agent reinforcement learning (MARL) provides a framework for modeling complex interactions among agents, with methods typically classified into policy-based, value decomposition, offline, communication-based, and exploration-based approaches.

Policy-based methods, such as COMA (Foerster et al., 2018), MADDPG (Lowe et al., 2017), and MAPPO (Yu et al., 2022), optimize agent policies using gradients. In contrast, value decomposition methods, include VDN (Sunehag et al., 2018), QMIX (Rashid et al., 2018), QPLEX (Wang et al., 2021a), QTRAN (Son et al., 2019), ResQ (Shen et al., 2022), RiskQ (Shen et al., 2023), and RMIX (Qiu et al., 2021), which are particularly effective in cooperative tasks.

Offline methods, such as MADIFF (Zhu et al., 2024) and DoF (Li et al., 2024a) using diffusion to generate trajectories and policies of agents. Communication-based approaches, such as CommNet (Sukhbaatar et al., 2016), TarMAC (Das et al., 2019), ToM2C (Wang et al., 2021b), and CommFormer (Hu et al., 2024), focus on enhancing multi-agent cooperation through improved communication. Exploration-based methods like MAVEN (Mahajan et al., 2019) and ICES (Li et al., 2024c) aim to strengthen agents' exploration capabilities to better adapt to dynamic and complex environments.

Our method, GradPS, is orthogonal to these techniques, focusing on improving learning efficiency by mitigating gradient conflicts among agents.

# C. Method

## C.1. Gradient Conflict

Gradient conflicts commonly occur in multi-agent systems. Even if agents share a common goal, they may have different partial observations, be located in different positions of the game which may lead to different agent gradients (e.g., divergent actions). For example, when capturing one prey, it is possible that the best action for agent 1 is to move left, while the best action for agent 2 is to move right. This will cause gradient conflict when the optimal actions differ between agents.

We apply VDN and QMIX to run 3m scenario. As shown in Figure 1, gradient phenomenon still exists when using different optimizers. The gradient conflict in multi-agent reinforcement learning is shown in Figure 2.

## C.2. Gradient Monitoring

The gradient of neurons refers to the gradient on the activation layer in a neural network. To calculate the gradient for each agent, we designed a gradient detection layer capable of computing the partial derivative of the loss with respect to the observation of a specific agent. Specifically, the input data to the detection layer has dimensions of $(batch \times agents, dim)$, and the layer's weights are a fixed all-ones matrix with dimensions $(agents, dim)$. The input dimensions are reshaped to

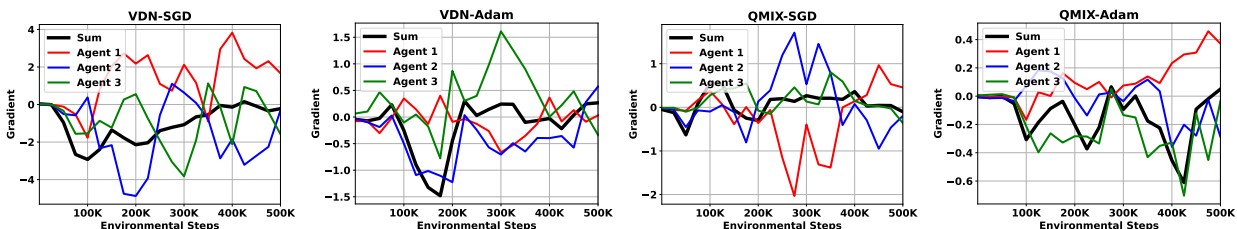

Figure 1: Gradient conflicts in VDN and QMIX when using different optimizers.

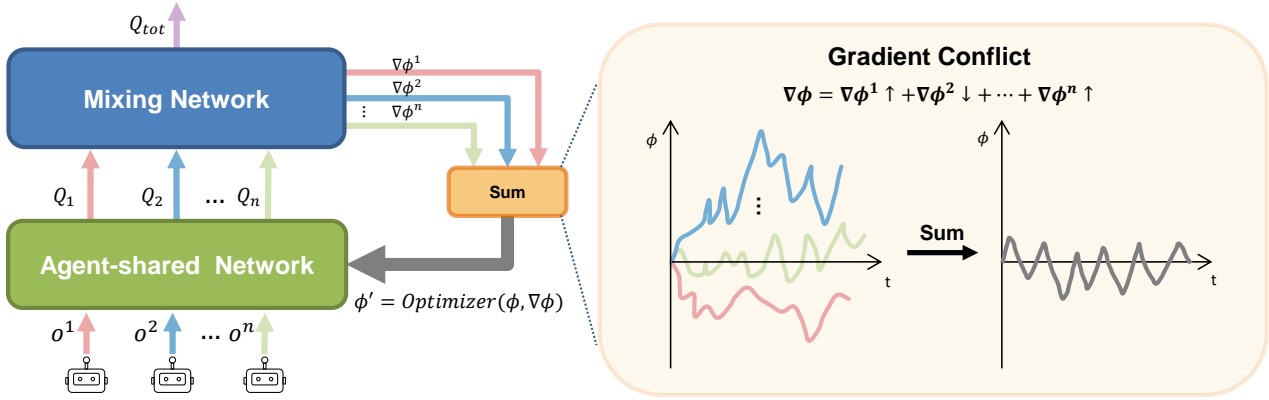

Figure 2: Gradient conflicts in multi-agent reinforcement learning (MARL). Different colors represent the observations and gradients of different agents. The overall gradient, computed by summing the gradients of all agents, is used to optimize the parameters of the agent network. However, conflicts may arise when positive and negative gradients counteract each other.

$(batch, agents, dim)$, then multiplied element-wise by the all-ones matrix. The result is subsequently reshaped back to the original dimensions of the input data to produce the output.

After the backpropagation of loss, each detection layer obtains the calculated partial derivative, which is the gradient of each neuron. We obtain the gradient of loss in the detection layer and clear these gradients to prevent the optimizer from updating the parameters of the detection layer. Afterward, the optimizer will continue to update the network parameters using the gradients of each parameter as usual.

This allows the gradient information to be retained in the gradient detection layer without altering the original data values. $Batch$ refers to the batch size, $agents$ represents the number of agents, and $dim$ denotes the number of neurons.

### C.3. Conflict-Based Grouping

The gradients for the most recent $T$ target network update periods are accumulated and recorded. Please refer to Appendix C.2 for a detailed procedure of gradient recording. At the end of each target network update period, we evaluate the neuron efficiency of neurons with $T$ period data and identify futile neurons if their neuron efficiency (defined in (4)) is low.

The appearance of futile neurons indicates significant conflicts in the agents during recent periods. To reduce gradient conflicts, We divide the agents into $K$ groups based on their gradient conflict values accumulated over $T$ periods, aiming to minimize the total gradient conflict within each group.

Based on the gradient conflict values between different agents, we apply the spectral clustering algorithm to clustering the agents. The clustering information is implicitly represented by the gradient conflict values between the agents. Therefore, we utilize the Gaussian kernel function to calculate the distance between agents based on gradient conflicts. First, the conflict matrix is transformed into an affinity matrix $A$ using the Gaussian kernel function. The affinity matrix represents the similarity between agents. Each cell $A_{ij}$ is defined as follows.

$$A_{ij} = e^{\left(-\frac{C_{ij}^2}{2\sigma^2}\right)} \tag{C.1}$$

where $C_{ij}$ is the Neuron Gradient Conflict between agent $i$ and $j$ defined in (3), and $\sigma$ is the scale parameter of the Gaussian

kernel, which controls the influence of distance (conflict value) on similarity. Typically, $\sigma$ is chosen to be close to the mean or median of the sample distances, ensuring that the affinity matrix is neither too sparse nor too dense.

The graph Laplacian matrix $L$ is computed as $L = D - A$, where $D$ is the degree matrix (the degree of each node is the sum of the weights of its connected edges) and $A$ is the affinity matrix.

$$D_{i,j} = \begin{cases} 0, & if \quad i \neq j \\ \sum_j A_{i,j}, & if \quad i = j \end{cases} \tag{C.2}$$

The Laplacian matrix is subjected to spectral decomposition, and the top $K$ eigenvectors corresponding to the smallest eigenvalues are selected as the embedding vectors. In the embedding space, K-means clustering is applied to these eigenvectors to partition the agents into $K$ groups.

### C.4. Algorithm

The detailed GradPS algorithm is described in Algorithm 2.

---

**Algorithm 2** GradPS

---

**Require:** cumulative gradient interval $T$, number of groups $K$, group restore probability $\rho$
 1: Initialize parameters of the network $\phi$
 2: Initialize parameters of the target network $\phi'$
 3: Initialize replay buffer $\mathcal{D}$
 4: Initialize Neural Network Optimizer
 5: Initialize the SGD optimizer for restoring
 6: **for** $e \in \{1, \ldots, m \text{ episodes}\}$ **do**
 7:     Start a new episode;
 8:     **while** Episode is not end **do**
 9:         Get the Agent action $a_i$
 10:         Execute $a_i$, obtain global reward $r$ and the next state $s'$
 11:         Update replay buffer $\mathcal{D}$
 12:         Sample a batch $\mathcal{D}'$ from replay buffer $\mathcal{D}$
 13:         $Loss(\theta, \phi) = (Q(s, a; \theta, \phi) - y_{s,a})^2$
 14:         Accumulated the gradient of each agent
 15:         Update $\phi$ by $Loss$
 16:     **end while**
 17:     **if** Target network update **then**
 18:         Get the set of neurons that have accumulated $T$ period gradients in the first activation layer
 19:         Select the futile neurons
 20:         **for** Each futile neuron **do**
 21:             Grouping agents into $K$ groups according to gradient conflicts
 22:             Make $K$ copies of the parameters
 23:             Assign parameters of the corresponding group to each agent
 24:             Clear the accumulated gradient of this neuron
 25:         **end for**
 26:         **if** Restore operation required **then**
 27:             Calculate the variance of each neuron
 28:             Set the SGD optimizer to start resetting the weights of this neuron
 29:         **end if**
 30:         **if** Some neurons have already been reset **then**
 31:             Restore these neurons to fully parameter sharing
 32:         **end if**
 33:         $\phi' = \phi$
 34:     **end if**
 35: **end for**

---

# D. Experimental Details

## D.1. Experimental Setup

We compare GradPS with six parameter sharing methods, including Kaleidoscope (Li et al., 2024d), Selective Parameter Sharing (SePS) (Christianos et al., 2021), Structured Network Pruning with parameter Sharing(SNP) (Kim & Sung, 2023), Parameter Sharing (FuPS), Full Parameter Sharing with index (FuPS+id), No Parameter Sharing (NoPS). For the sake of robustness, each experiment was carried out using different random seeds at least 5 times.

Table 1: Baseline parameter sharing algorithms

| Algorithms | Description |
|---|---|
| FuPS[1] [2] | Agents share all the parameters. |
| FuPS+id | Agents share all the parameters with agent ID in input. |
| SNP (Kim & Sung, 2023) | The network is randomly pruned subnetworks. |
| SePS[3] (Christianos et al., 2021) | Agents are clustered to share parameters within each cluster. |
| NoPS | Agents do not share any parameters. |
| Kaleidoscope[4] (Li et al., 2024d) | Agents share parameters based on learnable masks. |

We implement these algorithms based on their open-source repositories to carry out performance analyses with hyperparameters consistent with those in PyMARL. Our methods are implemented within the PyMARL framework, and each is evaluated using five random seeds with $95\%$ confidence intervals. Specific hyperparameters of different environments are listed in Table 2. We conduct experiments on a cluster equipped with multiple NVIDIA GeForce RTX 3090 GPUs.

Table 2: Hyperparameter of different environments

| Hyperparameter | SMAC | Predator-Prey |
|---|---|---|
| Action Selector | epsilon greedy | epsilon greedy |
| Batch Size | 32 | 32 |
| Buffer Size | 5000 | 5000 |
| Learning Rate | 0.0005 | 0.0005 |
| Hypernet Embed Dimension | 64 | 64 |
| Target Update Interval | 200 | 200 |
| Futile Threshold $\alpha$ | 0.2 | 0.2 |
| Accumulated Period $T$ | 20 | 10 |
| Group Number $K$ | 3 | 2 |
| Restore Probability $\rho$ | 0.05 | 0.025 |
| Reg Learning Rate | 0.005 | 0.005 |

## D.2. Environment

### D.2.1. PREDATOR-PREY

The predator-prey scenario simulates a grid world where multiple agents cooperate to capture prey scattered across the map. The prey consists of two types: stags and hares. Capturing a hare requires only one agent while capturing a stag necessitates at least two agents working together. Successfully capturing a hare provides a team reward of $+2$, while capturing a deer yields a reward of $+10$.

Some predators possess a special, unobservable trait that enables them to receive the same team reward for capturing a hare as they would for capturing a stag. The goal is to maximize the total team reward within a limited timeframe. We have developed three different environmental configurations—small, medium, and large—each with varying numbers of agents and prey, as well as different map sizes.

---

[1] https://github.com/oxwhirl/pymarl
[2] https://github.com/tjuHaoXiaotian/pymarl3
[3] https://github.com/uoe-agents/seps
[4] https://github.com/LXXXXR/Kaleidoscope

**Game Rules**

- **Agent Movement**: Agents can move in four directions or stay in place. Movement is restricted by the presence of other agents or preys.

- **Observation and Decision Making**: Each agent observes a 7x7 grid centered around itself, receiving information about nearby agents and preys. Decisions are based on this local observation.

- **Capture Mechanism**: To capture a stag, at least two agents must be adjacent to it and must choose the *capture* action at the same time. In contrast, capturing a hare requires only one agent. Successful capture relies on strategic positioning and synchronized actions among agents.

- **Rewards and Penalties**: Agents receive a positive reward for each prey captured through cooperative action.

- **Episode Termination**: An episode terminates if all preys are captured or after 150 steps, providing a fixed time frame for agents to maximize their collective reward.

| Configuration | Number of All Predators | Number of Special Predators | Number of Stags | Number of Hares | Map Size | Reward for Stags | Reward for Hares |
|---|---|---|---|---|---|---|---|
| **Small** | 6 | 2 | 6 | 6 | 20 x 20 | +10 | +2 |
| **Medium** | 10 | 3 | 10 | 10 | 25 x 25 | +10 | +2 |
| **Large** | 15 | 5 | 15 | 15 | 30 x 30 | +10 | +2 |

Table 3: Comparison of Predator-Prey Configurations

### D.2.2. STARCRAFT II MULTI-AGENT CHALLENGES (SMAC)

The StarCraft Multi-Agent Challenge (SMAC) (Samvelyan et al., 2019) is a popular benchmark used extensively in the domain of multi-agent reinforcement learning. Built on the StarCraft II game engine, SMAC specializes in micromanagement scenarios where each agent is controlled by an independent agent who must make decisions based on local observations. MARL algorithms coordinate a team of agents to engage in combat against an opposing team managed by the game's built-in AI.

The performance of these algorithms is quantitatively evaluated by the test win rate or the test return of the gameplay. Table 4 depicts the overview of SMAC scenarios used in the experiment.

| Name | Difficulty | Ally Units | Enemy Units |
|---|---|---|---|
| 3m | Easy | 3 Marines | 3 Marines |
| 5m | Easy | 5 Marines | 5 Marines |
| 15m | Easy | 15 Marines | 15 Marines |
| 9z | Easy | 9 Zealots | 9 Zealots |
| 3s6z | Easy | 3 Stalkers & 6 Zealots | 3 Stalkers & 6 Zealots |
| 1c3s5z | Easy | 1 Colossi & 3 Stalkers & 5 Zealots | 1 Colossi & 3 Stalkers & 5 Zealots |
| 5m_vs_6m | Hard | 5 Marines | 6 Marines |
| 2c_vs_64zg | Hard | 2 Colossi | 64 Zerglings |
| MMM2 | Super Hard | 1 Medivac, 2 Marauders & 7 Marines | 1 Medivac, 3 Marauders & 8 Marines |
| 27m_vs_30m | Super Hard | 27 Marines | 30 Marines |
| 1c3s8z_vs_1c3s9z | Super Hard | 1 Colossi & 3 Stalkers & 8 Zealots | 1 Colossi & 3 Stalkers & 9 Zealots |
| 3s5z_vs_3s6z | Super Hard | 3 Stalkers & 5 Zealots | 3 Stalkers & 6 Zealots |

Table 4: Overview of SMAC scenarios used in the experiment.

### D.2.3. One Step Matrix Game

The One-Step Matrix Game is a classic experimental environment used to evaluate the performance of multi-agent reinforcement learning (MARL) algorithms. By designing a simple reward structure in the form of a matrix, it simulates cooperative or competitive interactions among multiple agents, enabling the assessment of an algorithm's ability to handle nonlinear relationships and coordination challenges. In this study, the One-Step Matrix Game involves two agents, each selecting an action, where the combination of all agents' actions constitutes a joint action. The rows and columns of the matrix represent the action choices of the respective agents, and each element within the matrix denotes the reward value corresponding to a specific joint action.

| $u_1$ \ $u_2$ | $A$ | $B$ | $C$ | $D$ |
|---|---|---|---|---|
| $A$ | 0 | -1 | -2 | -3 |
| $B$ | 1 | 0 | -1 | -2 |
| $C$ | 2 | 1 | 0 | -1 |
| $D$ | 3 | 2 | 1 | 0 |

Table 5: $f_2 = -1 \times f_1$

| $u_1$ \ $u_2$ | $A$ | $B$ | $C$ | $D$ |
|---|---|---|---|---|
| $A$ | 0 | 0 | 0 | 0 |
| $B$ | 1 | 1 | 1 | 1 |
| $C$ | 2 | 2 | 2 | 2 |
| $D$ | 3 | 3 | 3 | 3 |

Table 6: $f_2 = 0 \times f_1$

| $u_1$ \ $u_2$ | $A$ | $B$ | $C$ | $D$ |
|---|---|---|---|---|
| $A$ | 0 | 0.5 | 1 | 1.5 |
| $B$ | 1 | 1.5 | 2 | 2.5 |
| $C$ | 2 | 2.5 | 3 | 3.5 |
| $D$ | 3 | 3.5 | 4 | 4.5 |

Table 7: $f_2 = 0.5 \times f_1$

| $u_1$ \ $u_2$ | $A$ | $B$ | $C$ | $D$ |
|---|---|---|---|---|
| $A$ | 0 | 1 | 2 | 3 |
| $B$ | 1 | 2 | 3 | 4 |
| $C$ | 2 | 3 | 4 | 5 |
| $D$ | 3 | 4 | 5 | 6 |

Table 8: $f_2 = 1 \times f_1$

Table 9: Simple one-step payoff matrix game for understanding the impact of futile neurons on network expressiveness.

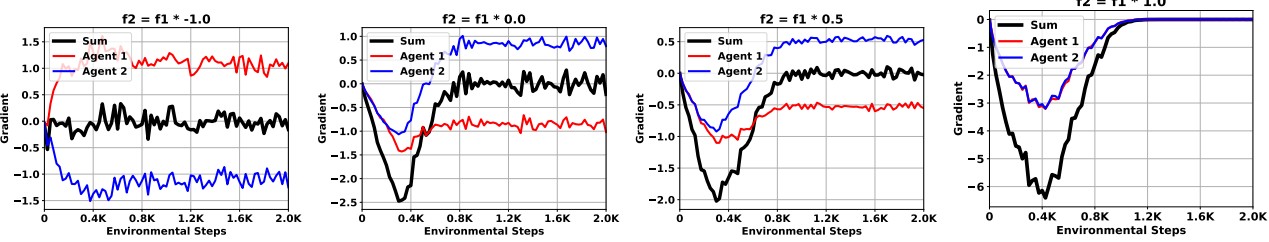

Figure 3: Conflicting neuron gradients in different one step payoff matrix games.

Figure 3 depicts the gradients for different agents in two payoff matrix game using the VDN methods. As it is depicted in the graph, the gradients for different agents can be significantly different. The conflict of gradients may lead to canceling out each other, which makes the sum of the gradient for one neuron close to zero. GradPS can further reduce bias by resolving gradient conflicts in neurons, as shown in Figure 4.

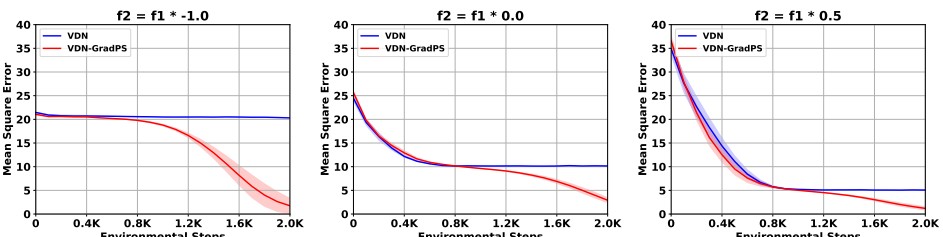

Figure 4: Applying GradPS to different one step payoff matrix games.

## D.3. Experimental Results

**Predator-Prey**

Figure 5 presents the return and the percentage of futile neurons in GradPS within the predator-prey environment, in comparison to other parameter-sharing algorithms.

**SMAC**

Figure 6 presents the win rate and the percentage of futile neurons in GradPS within the SMAC environment, in comparison to other parameter-sharing algorithms.

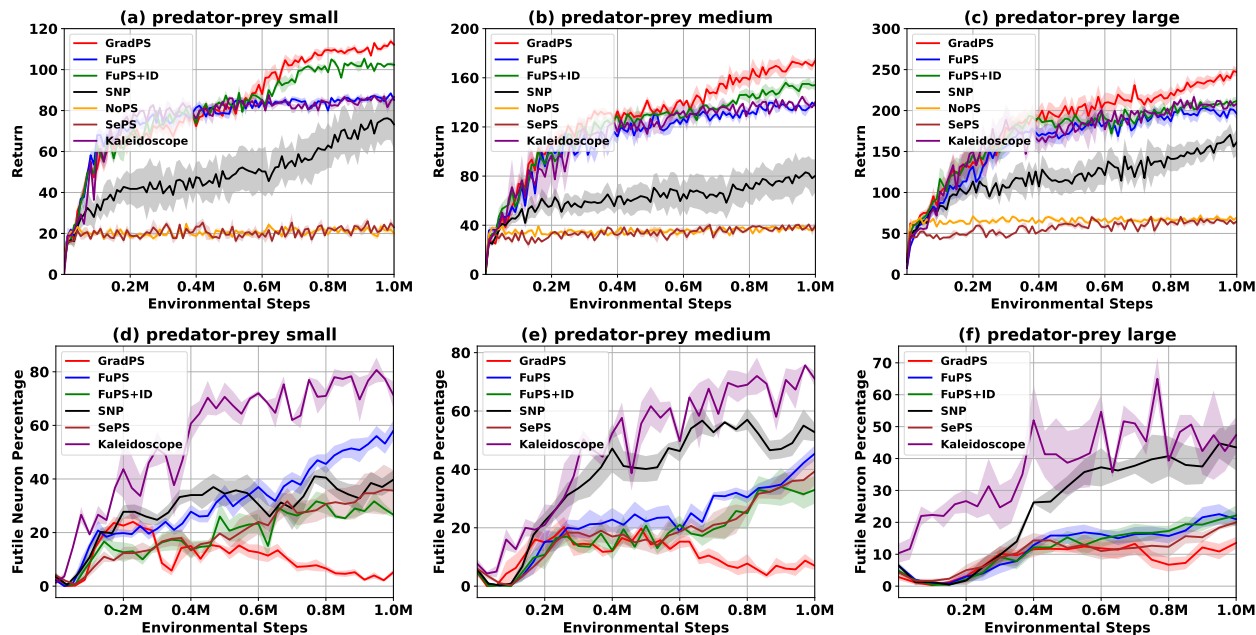

Figure 5: GradPS performs better than other PS methods on Predator-Prey: (a-c) the return for predator-prey small/medium/large environment. (d-f) the percentage of futile neurons for predator-prey small/medium/large environment.

## D.4. GradPS Performance Under PyMARL3 Settings

We extended the training time based on the settings of PyMARL and PyMARL3. The results are shown in Figure 7 and Figure 8.

## D.5. Ablation Study and Discussion

### D.5.1. COMPARISON OF FUTILE AND DORMANT NEURONS

| Feature | Dormant Neurons | Futile Neurons |
|---|---|---|
| Definition | Normalized Average Activation Value falls below | Neuron Efficiency is below |
| Normalized activation score | Small | Any |
| Gradient conflict | Unclear | Large |
| Gradient | Near Zero | Any |
| Parameter Update | Near Zero | Slowed down by gradient conflict |

Table 10: Overview of SMAC scenarios used in the experiment.

Dormant neurons (Sokar et al., 2023) have low normalized activation scores, while futile neurons suffer from gradient conflicts unrelated to activation. A non-dormant neuron can be a futile neuron. We have listed the differences as Table 10.

Additionally, we report the percentage of dormant neurons and futile neurons in agent networks with VDN during the training process in three simple payoff matrix games. Moreover, we evaluate the MSE of reconstructing the Payoff matrix when using ReDo (Sokar et al., 2023) and GradPS. ReDo and GradPS are methods developed to reduce the percentage of dormant neurons and futile neurons, respectively. The default parameters of ReDo and GradPS are used. The results are shown in Figure 9.

For the $f_2 = f_1 \cdot 0.0$ and $f_2 = f_1 \cdot 0.5$ games, the Neuron Percentage graphs show that dormant neurons gradually decrease and stabilize to fix values (around 10%), whereas futile neurons rise to 100% after a few thousand environment steps. For the $f_2 = f_1 \cdot -1.0$ game, the percents of the dormant neurons and the futile neurons fluctuate around 30%-40% and 80%-90%,

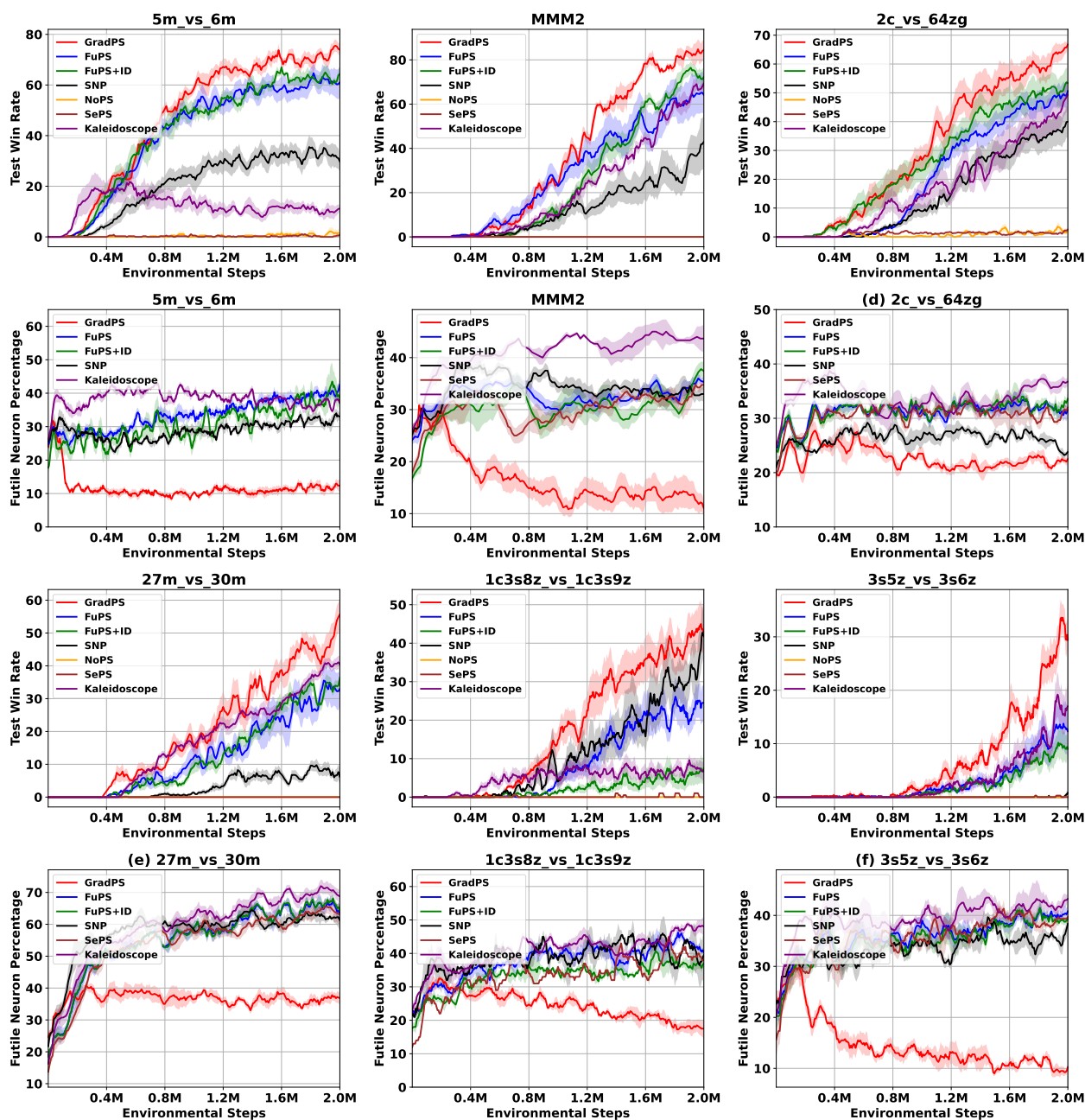

Figure 6: Comparison with other Parameter Sharing methods: the test win rate and the futile neuron percentage in SMAC

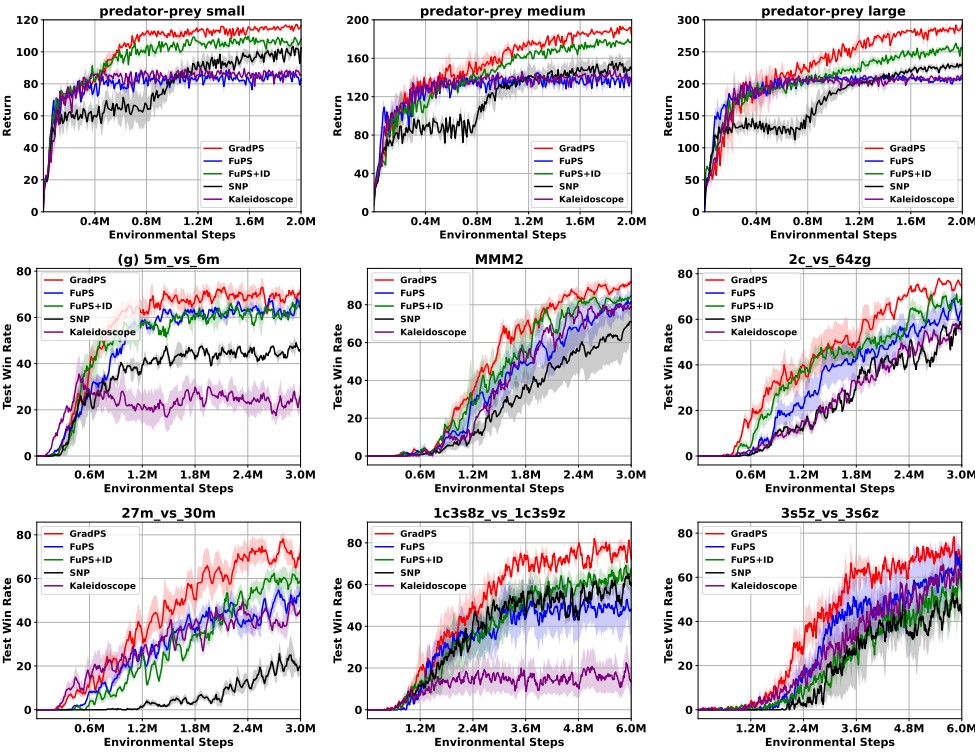

Figure 7: Comparison with other Parameter Sharing methods: the test win rate in SMAC and the return in Predator-Prey

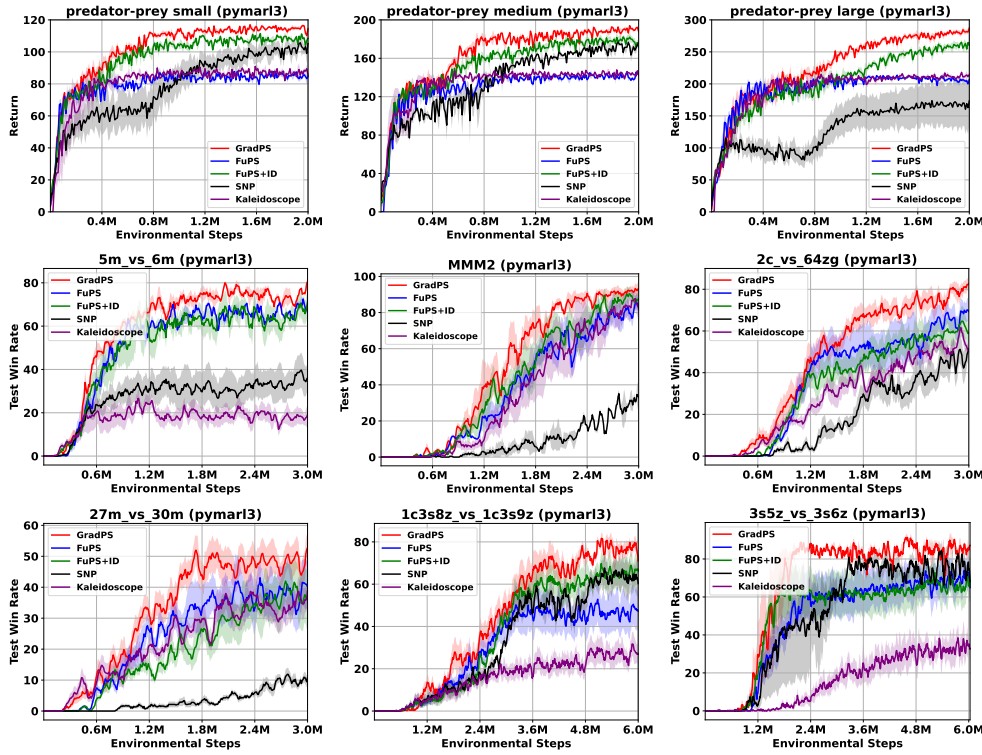

Figure 8: Comparison with other Parameter Sharing methods based on PyMARL3: the test win rate in SMAC and the return in Predator-Prey

respectively. These figures show that the percentages for the two types of neurons are different across training processes.

For all three matrix games, the MSE of reconstructed payoff matrix is shown in MSE.pdf figures. For these graphs, the MSE for the original VDN, VDN with ReDo (Sokar et al., 2023), VDN with GradPS are shown.

When using ReDo with VDN, the MSE does not change significantly compared to the VDN method. However, when using GradPS with VDN, the MSE drops significantly. For these matrix games, in terms of reducing MSE, GradPS, as a futile-neuron-method, works better than ReDo, a dormant-neuron-based method. These findings further highlight the significant difference among the two types of neurons.

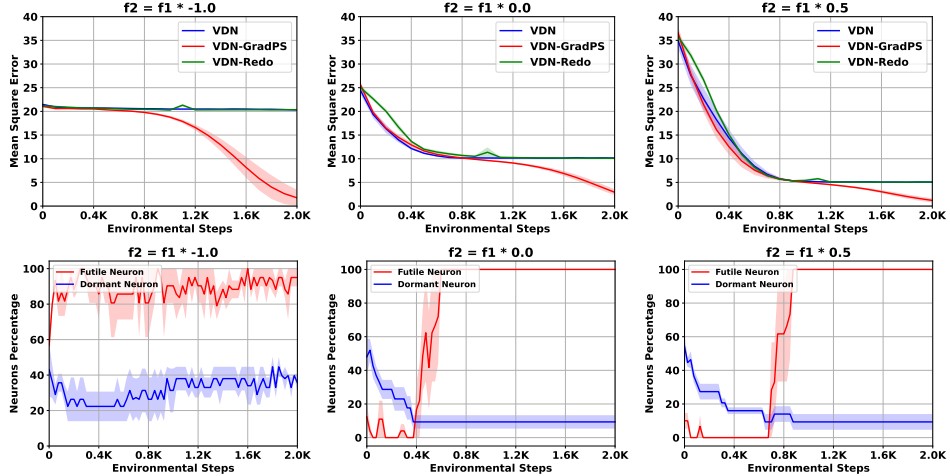

Figure 9: Conflicting neuron gradients in different one step payoff matrix games.

### D.5.2. THE IMPACT OF NETWORK WIDTH

In this section, we study whether gradient conflicts are different in networks of different widths. As shown in Figure 10, the percentage of futile neurons remains approximately equal across networks of different widths.

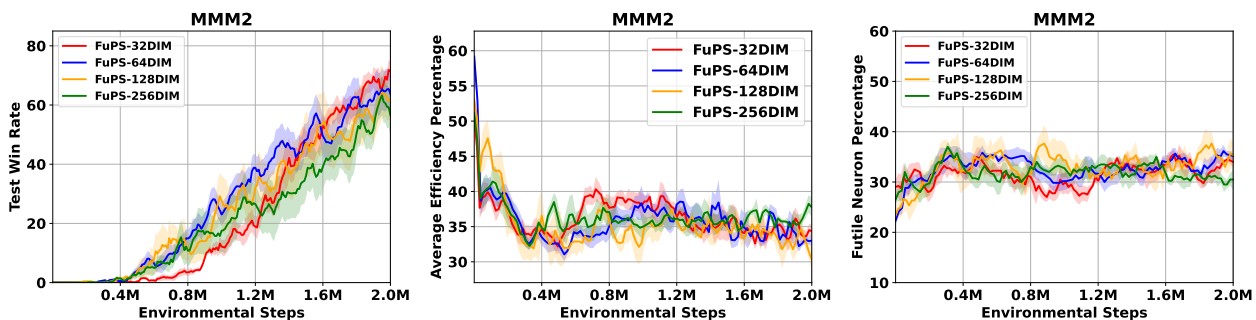

Figure 10: Average Efficiency Percentage and Futile Neurons Percentage in networks with different widths.

### D.5.3. SHARING AT DIFFERENT DEPTHS OF NETWORK LAYERS

In this section, we investigate whether gradient conflicts differ at different depths in the network. We deepen the QMIX agent network to have four linear-activation layers. We apply GradPS to these activation layers and detect gradient conflicts separately. The results are shown in Figure 11, the number of futile neurons in the first layer is greater than in the other layers.

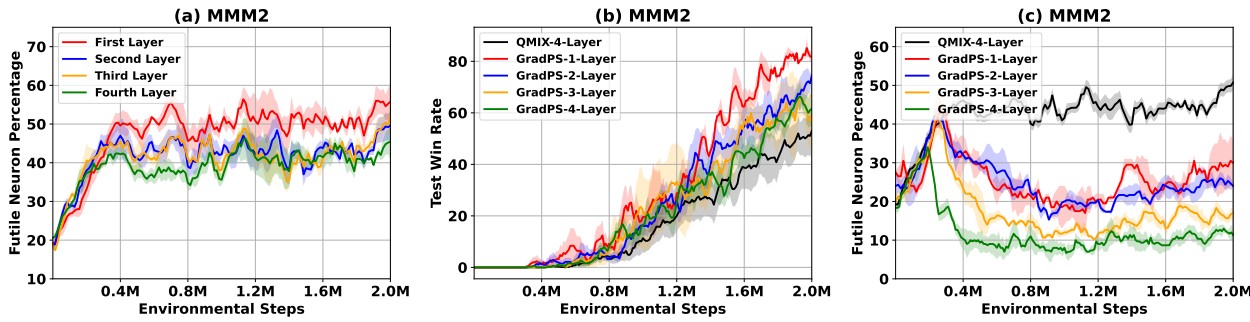

Figure 11: (a) The Futile Neurons Percentage at different depths of a four layer QMIX network. (b) The performance of GradPS in different network layers. (c) The Futile Neurons Percentage after using GradPS on different layers.

### D.6. Execution Time Table and Memory Usage Table.

We present the wall-clock training time (in hours) for each algorithm, averaged over 5 random seeds. The computational overhead of GradPS is small. It takes less training time than SNP, SePS and NoPS, most of the time.

Table 11: Training time of different parameter sharing algorithms

| Training Time(h) | FuPS | GradPS | FuPS+id | SNP | SePS | NoPS | Kaleidoscope |
|---|---|---|---|---|---|---|---|
| Predator-Prey Small | 2.45 | 2.74 | 2.55 | 2.72 | 5.23 | 5.45 | 4.37 |
| Predator-Prey Medium | 5.11 | 5.65 | 5.23 | 5.34 | 6.86 | 7.18 | 6.32 |
| Predator-Prey Large | 5.89 | 6.32 | 5.84 | 6.62 | 9.92 | 11.65 | 9.05 |
| 5m_vs_6m | 9.21 | 9.74 | 9.33 | 10.64 | 12.56 | 14.21 | 11.86 |
| MMM2 | 12.94 | 13.83 | 13.02 | 14.95 | 18.42 | 19.34 | 15.53 |
| 27m_vs_30m | 23.12 | 24.75 | 23.45 | 25.77 | 36.53 | 42.32 | 31.45 |

The following table depicts the number of parameters for the whole network and each agent network compared to the full parameter-sharing approach. The additional memory overhead of GradPS is lower than that of SePS and NoPS.

Table 12: Usage of the whole network parameters

| The whole network Parameter usage | FuPS | GradPS | FuPS+id | SNP | SePS | NoPS | Kaleidoscope |
|---|---|---|---|---|---|---|---|
| Predator-Prey small | 100% | 105% | 100% | 103% | 112% | 159% | 114% |
| Predator-Prey medium | 100% | 104% | 100% | 102% | 108% | 171% | 109% |
| Predator-Prey large | 100% | 103% | 100% | 101% | 105% | 178% | 107% |
| 5m_vs_6m | 100% | 116% | 101% | 106% | 180% | 261% | 151% |
| MMM2 | 100% | 122% | 101% | 108% | 156% | 354% | 134% |
| 27m_vs_30m | 100% | 114% | 101% | 105% | 128% | 466% | 117% |

Table 13: Usage of agent network parameters

| Agent network Parameter usage | FuPS | GradPS | FuPS+id | SNP | SePS | NoPS | Kaleidoscope |
|---|---|---|---|---|---|---|---|
| Predator-Prey Small | 100% | 145% | 102% | 123% | 198% | 624% | 218% |
| Predator-Prey Medium | 100% | 146% | 103% | 124% | 197% | 1034% | 217% |
| Predator-Prey Large | 100% | 147% | 104% | 124% | 196% | 1540% | 216% |
| 5m_vs_6m | 100% | 140% | 102% | 116% | 297% | 519% | 226% |
| MMM2 | 100% | 176% | 102% | 128% | 296% | 1036% | 218% |
| 27m_vs_30m | 100% | 195% | 105% | 136% | 293% | 2762% | 215% |

# E. Discussion

## E.1. Societal Impact

Our research primarily concentrates on the technical and theoretical aspects of multi-agent reinforcement learning, aiming to enhance the performance of these agents across a variety of tasks. While we do not foresee any direct negative consequences arising from our research, we are committed to maintaining an open dialogue. We highly appreciate and value constructive feedback from the community to ensure our work's contributions are beneficial and ethically sound.

## E.2. Limitations and Future Work

Although the gradient-based grouped parameter sharing method we proposed has achieved promising results compared to other approaches, there is still room for improvement. Our hyperparameters, such as the number of groups and the threshold for defining futile neurons, require further tuning. Additionally, increasing the number of groups may lead to higher parameter overhead. Defining futile neurons more effectively and exploring adaptive grouping will be key directions for our future research.

