# OpenReview forum: "GradPS: Resolving Futile Neurons in Parameter Sharing Network for Multi-Agent Reinforcement Learning"
_ICML.cc/2025/Conference — ICML 2025 poster_

### Official Review · Reviewer_P57Y · 2025-02-20

**Overall Recommendation:** 4

**Summary:**

This paper investigates parameter sharing techniques in cooperative multi-agent reinforcement learning (MARL). The authors observe gradient conflicts among multi-agent policies and propose a new partial parameter-sharing scheme. This method exhibits superior performance on benchmarks such as SMAC and PredatorPrey.

**Claims And Evidence:**

Yes.

**Essential References Not Discussed:**

No.

**Experimental Designs Or Analyses:**

Yes.

**Methods And Evaluation Criteria:**

1. The motivation is clear and well-supported by the toy example in Figures 1-3.
2. The experiments demonstrate the superior performance of the proposed GradPS compared to baselines on benchmarks.

Overall, the experiments are thorough enough to support the claims made in this paper. However, I would like to suggest a few additional experiments that could potentially strengthen the paper even further:
1. It might be insightful to visualize the group assignment pattern throughout the training process. This could offer valuable insights for future research on PS in MARL.
2. Considering that cloning and optimizing weights separately may introduce some overhead, it would be helpful to report the training time and memory usage.
3. \alpha also appears to be a quite important hyperparameter. it would be beneficial to include a parameter sensitivity analysis for this variable.

**Other Comments Or Suggestions:**

N/A

**Other Strengths And Weaknesses:**

This paper proposes a novel PS technique for MARL. The method is well-motivated by the observation that MARL often experiences gradient conflicts among agents, causing futile neurons and leading to degraded performance. The proposed method is clearly explained and well-supported by experiments. However, it might introduce extra overhead, and some experimental results may need further explanation (see the Question Section for details).

**Questions For Authors:**

Method-related:
1. How is gradient conflict retained during the interval T? Will this introduce significant overhead in terms of time and space?

Experiment-related:
1. From the analysis in Section 4, fewer futile neurons should lead to better expressiveness and performance. While the proposed GradPS provides the best performance and the lowest futile neuron percentage, baselines with higher futile neuron percentages do not seem to correspond to lower performance. Are there any possible reasons for this?
2. Could the authors provide a cost comparison (training time and memory requirements) for the proposed methods and baselines?
3. Is there any pattern or relationship between the clone ratio and the heterogeneity of the agents?
4. The authors state that \rho does not impact performance, though it seems to tie to the computation overhead. In this case, is it safe to assume that we should always set a large \rho?

**Relation To Broader Scientific Literature:**

As adequately discussed in the related work section, this work bridges the neuron learning efficiency and MARL.

**Theoretical Claims:**

N/A

---

> ### Author Rebuttal · Authors · 2025-04-01
>
> Thanks for your valuable comments and suggestions, your comments are very important to us. We will improve our work based on your suggestions. We address your concerns as follows.
>
>  ## Methods:
> >1. ... visualize the group assignment pattern...
>
> Thank you for your suggestion. It is very valuable, and we will include it in the revised manuscript. The following table shows an example of the grouping of **a neuron** in the 5m_vs_6m environment. Before 0.5M steps, the neuron is not futile. At 0.6M timesteps, the neuron is futile, and the agents for the neuron were grouped into (1,2,3) and (4,5). The grouping is restored at 1.2M timesteps. At 1.6M timesteps, the agents were grouped into (1,3) and (2,4,5).
>
> |Timesteps|0.2M|0.4M|0.6M|0.8M|1M|1.2M|1.4M|1.6M|1.8M|2M|
> |-|-|-|-|-|-|-|-|-|-|-|
> |Neuron|/|/|123/45|123/45|123/45|/|/|13/245|13/245|13/245|
>
> >2. ...report the training time and memory usage.
>
> Appendix D.6 shows the number of parameters (including factorization networks) used by different methods. We provide more results as follows.
>
> We present the wall-clock training time (in hours) for each algorithm, averaged over 5 random seeds. The computational overhead of GradPS is small. It takes less training time than SNP, SePS and NoPS, most of the time.
>
>
> |Time（h)|FuPS|GradPS|FuPS+id|SNP|SePS|NoPS|Kaleidoscope|
> |--|--|--|--|--|--|--|--|
> |Predator-Prey Small|2.45|2.74|2.55|2.72|5.23|5.45|4.37|
> |Predator-Prey Medium|5.11|5.65|5.23|5.34|6.86|7.18|6.32|
> |Predator-Prey Large|5.89|6.32|5.84|6.62|9.92|11.65|9.05|
> |5m_vs_6m|9.21|9.74|9.33|10.64|12.56|14.21|11.86|
> |MMM2|12.94|13.83|13.02|14.95|18.42|19.34|15.53|
> |27m_vs_30m|23.12|24.75|23.45|25.77|36.53|42.32|31.45|
>
> The following table depicts the number of parameters for each agent network compared to the full parameter-sharing approach. The additional memory overhead of GradPS is lower than that of SePS and NoPS.
>
> |Parameters|FuPS|GradPS|FuPS+id|SNP|SePS|NoPS|Kaleidoscope|
> |-|-|-|-|-|-|-|-|
> |Predator-Prey Small|100%|145%|102%|123%|198%|624%|218%|
> |Predator-Prey Medium|100%|146%|103%|124%|197%|1034%|217%|
> |Predator-Prey Large|100%|147%|104%|124%|196%|1540%|216%|
> |5m_vs_6m|100%|140%|102%|116%|297%|519%|226%|
> |MMM2|100%|176%|102%|128%|296%|1036%|218%|
> |27m_vs_30m|100%|195%|105%|136%|293%|2762%|215%|
>
>
> >3. ...$\alpha$ sensitivity analysis...
>
> $\alpha$ controls how futile neurons are identified. We have performed a sensitivity analysis for three environments; their results are depicted as follows.
>
>
> |Env.|$\alpha$=0.1|$\alpha$=0.2|$\alpha$=0.4|
> |-|-|-|-|
> |5m_vs_6m(win rate)|72.3|74.5|67.6|
> |MMM2(win rate)|78.3|82.6|67.6|
> |Predator-Prey Medium(return)|167.4|173.9|178.8|
>
> The results show setting $\alpha=0.2$ is a good choice. We agree that such a threshold may be environment-dependent, and we would like to explore it in the future.
>
> ## Questions:
> #### Method:
> > ...How gradient conflict retained..., introduce significant overhead...
>
> The accumulated gradients are stored in a tensor (T $\times$ agents $\times$ dim), representing the historical gradient values of neurons across each agent over the past T periods. These saved gradients are used to compute neuron efficiency, not for back-propagation. It introduces little computational overhead. Moreover, we have shown in the above parameter table that the memory overhead of GradPS is low.
>
> #### Experiment:
> >1. ...higher futile neuron percentages do not seem to correspond to lower performance. ...
>
> We appreciate your insight. MARL performance is influenced by multiple factors, and futile neuron is just one of them. We agree that higher neuron percentages do not necessarily lead to lower performance.
>
> >2. ... a cost comparison...
>
> We have already shown the execution time and memory overhead in the above table.
>
> >3. ...relationship between the clone ratio and the heterogeneity..
>
> The cloning ratio $K$ is not directly correlated with agent heterogeneity. $K$ is a neuron-level value, it is not an agent-level value. We will explore whether the aggregate $K$ of all the neurons can represent agent heterogeneity.
>
> >4. ...$\rho$  does not impact performance, though it seems to tie to the computation overhead...always set a large $\rho$
>
> $\rho$ is the probability of restoring the parameters of a neuron from grouped to shared status. We have conducted experiments that set $\rho$ to a large value (from 0.05 to 0.5), and we find that increased $\rho$ does not impact the computational overhead significantly.
>
> |Time(h)|$\rho$=0.05|$\rho$=0.2|$\rho$=0.5|
> |-|-|-|-|
> |5m_vs_6m|9.74|9.66|9.68|
> |MMM2|13.83|13.64|13.94|
>
> Gradual parameter recovery requires sufficient time to prevent rapid performance degradation; thus, a large $\rho$ may not be the optimal choice, as shown in the following table.
>
> |Env.|$\rho$=0.05|$\rho$=0.2|$\rho$=0.5|
> |-|-|-|-|
> |5m_vs_6m(win rate)|74.5|73.2|63.6|
> |MMM2(win rate)|82.6|80.2|57.6|

---

> > ### Comment · Reviewer_P57Y · 2025-04-02
> >
> > I appreciate the authors' rebuttal and the additional details provided. I encourage incorporating these updates into the revised manuscript to strengthen the paper. After careful consideration, I will maintain my original rating.

---

### Official Review · Reviewer_MPio · 2025-03-11

**Overall Recommendation:** 4

**Summary:**

This paper studies the gradient conflict in parameter-sharing networks and proposes a Gradient-based Parameter Sharing (GradPS) method to resolve futile neurons in the PS network. It dynamically creates multiple clones for each futile neuron. For each clone, a group of agents with low gradient-conflict shares the neuron’s parameters, which are updated according to the gradients of each agent group. GradPS performs better than the other parameters sharing (PS) methods for the SMAC and predator-prey benchmarks.

**Claims And Evidence:**

This paper is well-motivated. This paper shows the futile neuron phenomenon in MARL and the neuron gradient conflict of neurons by experiment on SMAC. I think the proposed method is interesting.

**Essential References Not Discussed:**

N/A

**Experimental Designs Or Analyses:**

This paper conducts evaluations on SMAC and predator-prey benchmarks. Experiments show that the proposed method outperforms the state-of-the-art PS methods. Ablation studies validate parameter sensitivity.

Some issues:

1. How do you determine the futile threshold $\alpha$ ?

2. I am concerned about the computational overhead of this method. While I noticed the Execution Time Table in the appendix, the paper does not provide any explanation for it.

**Methods And Evaluation Criteria:**

The proposed method GradPS addresses the challenge of gradient conflicts in parameter-sharing (PS) networks for MARL. By dynamically cloning futile neurons and grouping agents with low gradient conflicts, GradPS effectively balances the trade-off between sample efficiency and policy diversity.

Overall, the method and evaluation are well-designed.

**Other Comments Or Suggestions:**

N/A

**Other Strengths And Weaknesses:**

## Strengths
+ Originality: This paper studies the gradient conflict among agents, a concept not previously explored in depth in MARL. The proposed gradient-based PS method is novel and addresses the policy homogeneity in parameter-sharing networks from the angle of neuron gradient conflict among agents.
+ Significance: GradPS can learn diverse behaviors through multiple clones to avoid gradient conflict, and enjoys good sample efficiency by sharing the gradients among agents of the same clone neuron.
+ Clarity: The paper is well-structured and well-written.
## Weaknesses
+ Analysis of the futile neuron threshold $\alpha$ is missing.
+ The algorithm lacks sufficient theoretical analysis.

**Questions For Authors:**

+ Under what circumstances does gradient conflict usually occur? If agents share a common goal, such as pursuing the same prey, is there still gradient conflict or futile neuron?
+ Could you discuss situations where this approach is not effective or does not provide performance gains?

**Relation To Broader Scientific Literature:**

This paper tackles the issue of policy homogeneity in parameter-sharing networks for multi-agent reinforcement learning from the angle of neuron gradient conflict among agents. This paper highlights the futile neuron phenomenon and the neuron gradient conflict of neurons in MARL, which have been explored in multi-task learning but not in MARL previously.

**Theoretical Claims:**

N/A

---

> ### Author Rebuttal · Authors · 2025-04-01
>
> Thanks for viewing our work as interesting. Thanks for your valuable comments regarding the novelty of the futile neuron phenomenon and neuron gradient conflicts in MARL. We will improve our work based on your suggestions. We address your concerns as follows.
>
> ## Experimental:
>
> >1. How do you determine the futile threshold?
>
> We perform a simple parameter sensitivity experiment to determine the futile threshold $\alpha$. The experimental results for SMAC (5m_vs_6m and MMM2) and Predator-Prey are shown as follows.
>
> |Env.| $\alpha$=0.1 | $\alpha$=0.2 | $\alpha$=0.4
> |--|--|--|--|
> |5m_vs_6m(win rate)|72.3|74.5|67.6|
> |MMM2(win rate)|78.3|82.6|67.6|
> |Predator-Prey Medium(return)|167.4|173.9|178.8|
>
> The sensitivity experiment shows that the performance for $\alpha=0.2$ is good. Thus, we fix the futile threshold of all the experiments as 0.2. It is possible that such a threshold should be flexible for different environments. Moreover, better thresholds, such as deviation from the average futile ratio, could be used. We want to explore these in the future.
>
> >2. ...the computational overhead of this method...
>
> In the appendix, the execution time (wall-clock training time) is shown as a ratio over the FuPS approach. The experimental results show that the additional computational overhead brought by GradPS is small; it is only slightly higher than FuPS. The other methods (e.g., SePS, NoPS) require significant computational overhead.
>
> In the following table, we present the wall-clock training time (in hours) for each algorithm, averaged over five random seeds. GradPS achieves lower computational overhead than other parameter-sharing methods (SNP, NoPS).
>
> |Execution time（h)|FuPS|GradPS|FuPS+id|SNP|SePS|NoPS|Kaleidoscope|
> |--|--|--|--|--|--|--|--|
> |Predator-Prey Small(1M)|2.45|2.74|2.55|2.72|5.23|5.45|4.37|
> |Predator-Prey Medium(1M)|5.11|5.65|5.23|5.34|6.86|7.18|6.32|
> |Predator-Prey Large(1M)|5.89|6.32|5.84|6.62|9.92|11.65|9.05|
> |5m_vs_6m(2M)|9.21|9.74|9.33|10.64|12.56|14.21|11.86|
> |MMM2(2M)|12.94|13.83|13.02|14.95|18.42|19.34|15.53|
> |27m_vs_30m(2M)|23.12|24.75|23.45|25.77|36.53|42.32|31.45|
>
> ## Weaknesses:
>
> >1.Analysis of the futile neuron threshold $\alpha$ is missing.
>
> We have discussed $\alpha$ in the above response.
>
> >2. The algorithm lacks sufficient theoretical analysis.
>
> We employ a simple yet effective method to mitigate gradient conflicts while encouraging future research to develop more solutions to gradient conflict issues in MARL. We will perform an in-depth theoretical analysis of GradPS and other parameter-sharing methods.
>
> ## Suggestions
>
> >1. Under what circumstances does gradient conflict usually occur? If agents share a common goal...
>
> Thanks for raising this question; we will discuss it in the discussion section. Gradient conflicts commonly occur in multi-agent systems. Even if agents share a common goal, they may have different partial observations, be located in different positions of the game, and their actions may be different, which may lead to different agent gradients (e.g., divergent actions). For example, when capturing one prey, it is possible that the best action for agent 1 is to move up, while the best action for agent 2 is to move down. The optimal actions for these two agents are different; this will cause gradient conflict, too.
>
> To verify the above analysis, we train two predators to capture one prey in a 5 $\times$ 5 map, and the spawn point of the agents and the prey is fixed. At each time, there is only one prey. Experiments show that the proportion of futile neurons is about 20%, which suggests that there are gradient conflicts, even if the two agents are pursuing the same prey.
>
> >2...situations where this approach is not effective...
>
> In a highly stochastic environment, SMACv2, where agent types are randomized each episode, GradPS fails to deliver performance improvements as the gradient conflict patterns vary due to the randomness of agents themselves.

---

> > ### Comment · Reviewer_MPio · 2025-04-06
> >
> > Thank you for the rebuttal and the additional details provided. My concerns have been addressed, and I will maintain my original rating.

---

### Official Review · Reviewer_EAb8 · 2025-03-13

**Overall Recommendation:** 2

**Summary:**

This paper addresses the balance between parameter sharing and behavioral diversity in MARL from the perspective of gradient conflict and resolution. First, the concept of gradient conflict in MARL is introduced, and experiments verify its impact and patterns on multi-agent policy training. Subsequently, a method is proposed to group agents at the neuron level by detecting gradient conflicts. Finally, the effectiveness of the proposed method is validated in SMAC and particle-based environments.

**Claims And Evidence:**

The paper's structure is generally sound, supported by relevant evidence. However, some arguments may lack rigor or contain inaccuracies. These are pointed out below in order of appearance:

1. （minor）Line 40 claims the work is "neuron-based" to distinguish it from others. This may be inaccurate, as other works (e.g., network pruning in PS) can also be considered neuron-based.

2. （major）The paper makes multiple direct connections between an agent's policy/acquired skills and its gradients. This connection may be problematic. Gradients indicate only a temporary direction of improvement for the neural network. First, this improvement might not align with the optimal direction. Furthermore, a momentary gradient direction does not immediately constitute a stable policy or skill.

3. (minor) Lines 181-184 state that neuron efficiency decreases with an increasing number of agents. This is clearly not a universal conclusion. In MARL scenarios where agents perform identical “tasks”, increasing the number of agents might actually increase neuron efficiency.

**Essential References Not Discussed:**

Since the abstract and introduction both mention the relationship between parameter sharing and diversity learning, the related work section should also include works on diversity-based learning. Although Section 3.1 already covers some diversity-based MARL works, some are still missing. There is a considerable amount of work in this area, including but not limited to: CDS [1], RODE [2], LIPO [3], MADP [4], FOX [5], and DICO [6].

[1] Celebrating diversity in shared multi-agent reinforcement learning. NeurIPS 2021.

[2] Rode: Learning roles to decompose multi-agent tasks.ICLR 2021.

[3] Generating diverse cooperative agents by learning incompatible policies.ICLR 2023.

[4] Measuring Policy Distance for Multi-Agent Reinforcement Learning. AAMAS 2024.

[5] Fox: Formation-aware exploration in multi-agent reinforcement learning. AAAI 2024.

[6] Controlling behavioral diversity in multi-agent reinforcement learning.ICML 2024

**Experimental Designs Or Analyses:**

The overall design of the experimental section in the paper is reasonable, but there are some issues with its specific implementation.

（Minor）Regarding the choice of baselines, I appreciate the authors' selection of SePS, SNP, and Kaleidoscope for comparison. However, considering that the proposed method performs dynamic grouping, it would be beneficial to also compare it with other dynamic grouping methods. Examples include AdaPS, ADMIN [1], or MADPS [2], which are mentioned in the Related Work section.

（Minor）Regarding the ablation study, since the authors mention a specially designed parameter restoration approach in lines 302-308, this approach should be compared with other restoration methods.

[1] ADMN: Agent-Driven Modular Network for Dynamic Parameter Sharing
in Cooperative Multi-Agent Reinforcement Learning. IJCAI 2024.

[2] Measuring Policy Distance for Multi-Agent Reinforcement Learning. AAMAS 2024.

**Methods And Evaluation Criteria:**

This paper constructs a strong PS algorithm from the perspective of gradient conflict and resolution. The overall idea is suitable for this problem. However, concerns about the proposed method's necessity and practicality are the primary reasons I cannot give this paper a high score.

(Major) Regarding necessity, the proposed method uses gradient conflict to identify operable neurons and divide agents. However, I believe it does not address the core issue. This method dynamically assigns “roles” to agents by identifying gradient conflicts, but the essence should be to identify the functional heterogeneity of agents in MARL tasks.

(Major) Regarding practicality, the proposed method still relies on two hyperparameters: the number of groups and the threshold for detecting gradient conflicts. This even introduces more hyperparameters compared to existing grouping methods (e.g., SePS), which significantly hinders its practical application.

**Other Comments Or Suggestions:**

Considering the concerns regarding necessity and practicality, I recommend that the authors emphasize investigating the fundamental connection between gradient-conflicted neurons and essential MARL elements, instead of concentrating solely on presenting a new algorithm.

**Other Strengths And Weaknesses:**

**Other Strengths**:

1. The paper investigates diversity learning in MARL from a novel perspective: gradient conflict.
2. The paper proposes a neuron-level parameter allocation method.

**Other Weaknesses**:

Practicality: Besides the aforementioned concerns about hyperparameter usage, I have concerns about the computational cost and practicality of neuron-level agent assignment. The method requires calculating an assignment matrix of at least *Agent\*HiddenDim* size, which can be computationally expensive. My careful review of the code revealed that the agents' policies are restricted to shallow networks. Its effectiveness and computational load in deeper networks, which are common in real-world scenarios, warrant further investigation and may be limited.

**Questions For Authors:**

Beyond the specific concerns already raised, I have the following questions, which may influence my assessment of whether my potential concerns become definitive shortcomings.

1. Regarding the gradient calculation for individual neurons across different agents in Figure 1, I have several questions.
Shouldn't the backpropagated gradient during neuron training be a tensor, rather than a scalar positive or negative value?
Can the sign of a single neuron's gradient represent the "direction" of overall knowledge?
The overall knowledge should be a non-linear combination of all neurons, so,  wouldn't a holistic view of all neurons better represent the knowledge?

2. Regarding lines 138-140 ("These findings suggest that although the knowledge learned by agents may be diverse, their differences may cause different gradients for neurons."), what is the precise meaning of this statement?

3. In lines 274-->236 ("Gradient conflicts arise from the diverse observations and actions of individual agents.") Observations and actions are inputs and outputs of the network. However, gradients depend not only on network input/output structure but also on the loss function. I recommend a more thorough derivation, as this information should also encompass elements of the MDP, such as state transitions.

**Relation To Broader Scientific Literature:**

This paper offers a novel perspective on diversity learning in MARL, utilizing the concepts of gradient conflict and futile neurons.

**Theoretical Claims:**

This paper does not involve complex theoretical proofs. I have carefully reviewed all theoretical definitions and related claims.

I believe there are just some minor issues with the notation used in the formulas and definitions.

 For example, in Equations 1 and 2, the definitions of *gradient conflict* and *Neuron Efficiency* lack a subscript for neuron $n$. In Equation 3, the superscript in the definition of *Futile Neuron* could be easily mistaken for an exponent symbol.

---

> ### Author Rebuttal · Authors · 2025-04-01
>
> Thanks for your valuable comments, we address your concerns as follows.
> ## Claims:
> >1. .."neuron-based"..
>
> GradPS is a neuron-based method **from the angle of gradient conflict**. We agree that pruning-based PS can be viewed as neuron-based too.
>
> >2. ..a momentary gradient direction..
>
> We agree that gradients indicate temporary improvements. We will soften the claim regarding the direct connections between gradient and diversity/skill.
>
> >3. ..neuron efficiency decreases
>
> In Dec-POMDPs, agents with identical tasks may take different actions due to partial observations, leading to gradient conflicts from divergent optimal actions.
>
> In Figure 3 (Left), these agents share the same goal (removing all enemies) and with the same type. Neuron efficiency decreases with more agents due to higher partial observation diversity. Please refer to our response to Suggestion 1 of Reviewer MPio for experiments regarding agents with identical task.
>
> ## Methods:
> >1. ..the functional heterogeneity of agents..
>
> The relationship between gradient conflicts and functional differences is worth exploring. Even agents with an identical task can lead to gradient conflicts due to partial observation issues, as analyzed above. **Thus, analyzing the learning efficiency of MARL for gradient conflict is needed.**
>
> We do not link gradient conflicts with role (functional) differences. However, according to our experiments (Fig. 10), our approach is able to discover hidden roles.
>
> >2. .. more hyperparameters..hinders its practical application.
>
> Our work is practical, requiring (only 4) no more hyperparameters than existing grouping methods without the need of pre-training (unlike SePS/AdaPS). The table below compares hyperparameters across PS/grouping methods.
> |Method|Hyperparameter Count|Description|
> |-|-|-|
> |SePS|4|Clusters count, VAE training step, latent dimension, KL weight|
> |AdaPS|4|Clusters count, Drop threshold, VAE training trajectory, KL weight|
> |ADMN|4|Module count, Layer number, Module outputs size, Combination weight|
> |MADPS|5|Obs subset size, sampling times, Fusion/Division threshold, Fusion/division interval|
> |SNP|7|Actor/Critic Pruning ratios $\times$ 3, Subnetworks number|
> |Kaleidoscope|6|Reset probability, Ensembles, Actor/Critic diversity, Actor/Critic reset interval|
> ## Theoretical Claims:
> > ...formulas and definitions...
>
> We will fix the notations.
>
> ## Experimental:
> >1. ..compare with other grouping methods.. AdaPS, ADMN [1], or MADPS [2]
>
> ADMN and MADPS are **not open-source**. Reproduction attempts at such a short time would risk introducing unfair biases. GradPS performs better than AdaPS as shown in the following table.
> |Env|GradPS|AdaPS|
> |--|--|--|
> |Predator-Prey Medium|173.9|94.7|
> |5m_vs_6m|74.5|42.9|
> |MMM2|82.6|52.2|
> >2. ..compared with other restoration methods..
>
> We compare two restoration methods, which use the average value of the group or the value of a random clone to directly replace the neuron weights. Our method is better.
> |Map/GradPS|Our|Average Restoration|Random Restoration|
> |--|--|--|--|
> |Predator-Prey Medium|173.9|154.2|142.1|
> |5m_vs_6m|74.5|64.2|54.7|
> |MMM2|82.6|53.5|35.7|
> ## Essential References:
> >..some diversity-based MARL works..
>
> Thanks, we will discuss all of them in related work.
>
> ## Weaknesses:
> >.. computational cost and practicality.. effectiveness in deeper networks.
>
>
> We have shown in Appendix Table 10 and 11 that the increased overhead is small. It is much smaller than partial PS methods (e.g., SePS and NoPS).
>
> Our work is practical for multi-layer agents. In Appendix D.5.2, for a 4-linear-layer agent, there are more futile neurons in the first linear layer than the other layers. Although there are more layers, such a method (QMIX-4-layer) performs weaker than standard QMIX, due to futile neurons. Applying GradPS in the first linear layer leads to higher performance gain than applying it to the other layers, justifying our first-layer focus.
>
> Moreover, we show that GradPS works in different agent networks (distributional or risk-sensitive) in Appendix D.4. Figure 7.
>
>
> ## Suggestions:
> >..gradient-conflicted neurons and MARL elements..
>
> It is necessary to study the fundamental connection between futile neurons and essential MARL elements following ADMN and MADPS.
>
> ## Questions:
> >1. ..gradient be a tensor..the overall knowledge..
>
> Neuron gradient is not a vector (tensor) gradient, as defined in Definition 1. The neuron gradient is the gradient with respect to the output for a neuron, which is just a scalar (positive or negative). A holistic view of all neurons can better represent the whole knowledge.
>
> >2. ..the precise meaning..
>
> We will soften the claim regarding the direct relationship between gradient conflicts and policy diversity.
>
> >3. ..gradients depend not only on network input/output structure..
>
> Gradient conflicts correlate with diverse agent observations/actions and are influenced by factors such as loss functions and environmental stochasticity.

---

### Official Review · Reviewer_EmSH · 2025-03-22

**Overall Recommendation:** 1

**Summary:**

This paper identifies "futile neurons," neurons with conflicting gradient updates, in parameter-shared multi-agent reinforcement learning (MARL). It proposes GradPS, which dynamically clones these neurons, grouping agents with low gradient conflict to promote diversity and efficiency. Experiments on SMAC and Predator-Prey benchmarks show improved performance compared to existing parameter-sharing methods.

**Claims And Evidence:**

The main claims of improved diversity and performance are generally supported by experiments. However, convergence plots are missing, making it hard to assess long-term stability. More explicit reproducibility analysis (e.g., variability across random seeds) would also help validate the claims.

**Essential References Not Discussed:**

No.

**Experimental Designs Or Analyses:**

Experiments cover multiple relevant benchmarks. Still, convergence properties are not shown explicitly, and some analyses (Sections 5.2/5.3) lack clarity. Specifically, the grouping mechanism for neurons and sensitivity to hyperparameters (e.g., number of groups K) need clearer explanation.

**Methods And Evaluation Criteria:**

The proposed GradPS algorithm is innovative and appropriate for the MARL setting. Experiments involve relevant benchmarks and comparisons. However, it is unclear whether baseline methods are implemented optimally, such as from existing strong repositories, potentially biasing results in favor of GradPS.

**Other Comments Or Suggestions:**

Put more details in main paper, especially in the method section.

**Other Strengths And Weaknesses:**

Weaknesses:

Clarity:
1. Section 5.2 and 5.3 is hard to read.

Method:
1. The "futile" neuron seems a kind of dormant neuron. The necessity of this new definition is not convincing
2. The grouping method which is important is not mentioned in the main paper.


Experiment:
1. The performance after convergence is missed.
2. Lack of comparison with naive partial parameter-sharing baselines, e.g., the agents share the whole network but the last layer.

**Questions For Authors:**

1. Are all algorithms implemented based on the same optimized repo such as pymarl2[1] and pymarl[3]?
2. Is the introduction of "futile" necessary?
3. Why does the execution time of GradPS increase, since the number of parameters for each agent is the same as the fullps baseline?



[1] Hu, Jian, et al. "Rethinking the implementation tricks and monotonicity constraint in cooperative multi-agent reinforcement learning." arXiv preprint arXiv:2102.03479 (2021).

[2] Jianye, H. A. O., et al. "Boosting multiagent reinforcement learning via permutation invariant and permutation equivariant networks." The Eleventh International Conference on Learning Representations. 2022.

**Relation To Broader Scientific Literature:**

Related to MARL and deep learning phenomenon.

**Theoretical Claims:**

The paper does not make theoretical claims requiring proof validation. Definitions of gradient conflict and futile neurons are clearly presented and intuitive.

---

> ### Author Rebuttal · Authors · 2025-04-01
>
> ## Claims:
> > 1. ...convergence plots are missing, ...More explicit reproducibility analysis...
>
> All experiments were repeated with **5 different random seeds** to ensure reproducibility, with means and variances shown in all experimental result figures of this work.
>
> Convergence of GradPS are evaluated by running GradPS in three environments for 3M instead of 2M steps. The results are depicted as follows. After 2M steps, for MMM2, the performance of GradPS continues to increase; for Predator-Prey and 5m_vs_6m, their performance is stable.
>
> |Env.|0.5M|1M|1.5M|2M|2.5M|3M|
> |-|-|-|-|-|-|-|
> |Predator-Prey Medium(return)|138.5|174.4|187.5|191.2|189.7|190.4|191.5|
> |5m_vs_6m(win rate)|28.2|57.9|68.7|73.4|74.3|72.5|74.6|
> |MMM2(win rate)|4.8|25.1|67.5|79.3|85.2|91.4|90.2|
> ## Methods:
> >1. ...whether baseline methods are implemented optimally...
>
> For fairness, all parameter-sharing (PS) methods (e.g., FuPS and SNP) were implemented in the same repositories (e.g., pymarl).  PS implementation for pymarl (located in rnn_agent.py) is the same as pymarl2 and pymarl3. From this point, pymarl2 and pymarl3 are not optimized pymarl. Moreover, we also evaluate distributional and risk-sensitive agents, where are not included in pymarl2&3.
>
> We show in the following table that for pymarl3 QMIX experiments, GradPS outperformed FuPS+ID and FuPS. This shows that our method works in pymarl3.
>
> |Env (pymarl3)|GradPS|FuPS+Id|FuPS|
> |-|-|-|-|
> |Predator-Prey Medium (return)|187.2|165.8|139.6|
> |5m_vs_6m (win rate)|73.4|68.9|65.7|
> |MMM2 (win rate)|85.8|74.2|72.3|
> ## Experimental:
> >2 ...the grouping mechanism and sensitivity to hyperparameters...
>
> **We have described the grouping method in Section 5.1** and provided more details in Appendix C. We describe the hyperparameters and present more results as follows.
>
> $T$ is the period for identifying futile neurons. An excessively large T may delay futile neuron detection.
>
> $K$ denotes the number of groups. Setting K too large will lead to low sample efficiency.
>
> $\rho$ indicates the recovery probability for a group. In short-term tasks, even $\rho$=0 has a negligible effect on performance.
>
> Sensitivity analysis for $T$, $K$, and $\rho$ is shown in the left, middle, and right of Figure 10 in Appendix D.5.3, respectively.
>
> The $\alpha$ determines futile neuron identification. Our sensitivity analysis (see table) shows $\alpha$=0.2 yields promising results.
>
> |Env.| $\alpha$=0.1 | $\alpha$=0.2 | $\alpha$=0.4
> |--|--|--|--|
> |5m_vs_6m(win rate)|72.3|74.5|67.6|
> |MMM2(win rate)|78.3|82.6|67.6|
> |Predator-Prey Medium(return)|167.4|173.9|178.8|
> ## Weaknesses:
> >1. Section 5.2 and 5.3 is hard to read.
>
> We will revise them to increase readability.
>
> ### Method:
> >1. ...seems a kind of dormant neuron...
>
> Dormant neurons have low normalized activation scores, while futile neurons suffer from gradient conflicts unrelated to activation. A non-dormant neuron can be a futile neuron.
>
> We have listed the differences as follows.
>
> | Feature|Dormant Neuron|Futile Neuron|
> |-|-|-|
> |Definition|Normalized Average Activation Value falls below $\tau$ |Neuron Efficiency is below $\alpha$|
> |Normalized activation score|Small|Any|
> |Gradient conflict|Unclear|Large|
> |Gradient|Near Zero|Any|
> |Parameter Update|Near Zero|Slowed down by gradient conflict|
>
> >2. ...the grouping method is not mentioned in the main paper...
>
> **We have described the grouping method in Sec 5.1 of the main paper.**
>
> ### Experiment:
> >2. ...Lack of comparison with naive baselines...
>
> We compared with **SOTA partial PS methods**: SNP, which prunes a shared network for individual policies, and Kaleidoscope, which learns agent-specific masks. Our method outperforms both, as shown in experiments.
>
> Following the reviewer's suggestion, we also compared with Naive LastPS (sharing all but the last layer), where our approach demonstrates significantly better performance, as it is depicted in following table.
>
> |Env|GradPS|Naive LastPS|
> |-|-|-|
> |Predator-Prey Medium(return)|173.9|74.7|
> |5m_vs_6m(win rate)|74.5|54.6|
> |MMM2(win rate)|82.6|16.2|
> ## Suggestions:
> >Put more details in the method section.
>
> We will add more details to improve its readability.
>
> ## Questions:
> >2. Is the introduction of "futile" necessary?
>
> In PS networks, gradient conflicts produce "futile neurons" - neurons hindered by conflicting updates. As detailed in our comparison table, these are systematically identifiable through their distinct characteristics.
>
> >3. why...the execution time of GradPS increase..., is the same as the fullps baseline...
>
> In GradPS, the number of parameters per agent differs from that in FuPS, as demonstrated in Appendix D.6. GradPS requires additional parameters for cloning neurons and storing gradients. For computational overhead, GradPS requires about 1.1$\times$ the training time of FuPS; it requires significantly less (40-50%) time than SNP and NoPS.

---

> > ### Comment · Reviewer_EmSH · 2025-04-05
> >
> > Thank you for your response. However, the issue I raised has not been fully addressed.
> >
> > ## Performance after Convergence
> >
> > Authors add some experimental results of GradPS in the rebuttal. But apparently, most of the experiments in both the main paper and appendix are not converged, comparing convergence performance of baselines and GradPS makes the improvement convincing.
> >
> > ## Code Base
> >
> > I do not agree that "pymarl2 and pymarl3 are not optimized pymarl". And the authors do not answer my questions about whether all the algorithms are implemented fairly based on the same optimized code base.
> >
> > ## The Necessity of "Futile"
> >
> > Theoretical and experimental results are needed to show difference and relation between "dormant" and "futile".
> >
> > ## Presentation of Grouping Method
> >
> > My original review means the grouping method should not be present in only several purely text sentences with any equations.
> >
> > Overall, the experiments are not convincing and the introduction of "futile" is not fully supported for me. But this work is still interesting. The authors could consider making their experiments solid and add support for introducing new concepts.

---

> > > ### Author Response · Authors · 2025-04-09
> > >
> > > Thanks for your comments. We would like to point out that the review process is overseen by PC/SAC/AC. Thanks to the ICML 25 policy, we will release the reviews of this work regardless of whether this work is accepted or not.
> > >
> > > ## Summary of New Experiments
> > >
> > > 1. We compare GradPS and other PS methods with much longer training time to see convergence plots on nine sets of experiments.
> > > 2. We implement GradPS and other PS methods on Pymarl3, and conduct more than six sets of experiments.
> > > 3. We empirically evaluate dormant and futile neurons through three sets of experiments.
> > > 4. We compare GradPS with more PS baselines.
> > >
> > > ## Convergence Plot
> > >
> > > In the initial rebuttal, we had conducted multiple sets of experiments by running the SMAC environments for more than 2 million steps, and the predator prey environments for more than 1 million steps.
> > >
> > > Here, we have conducted **nine sets of experiments** with extended training steps. The results, shown in the Pymarl folder of the [anonymous link](https://anonymous.4open.science/r/review1_1-4871/), demonstrate that GradPS outperforms other methods after all algorithms converge.
> > >
> > > ## Code Base
> > >
> > > We have already responded directly in the previous rebuttal that all algorithms in the paper are implemented fairly based on the same repository (Pymarl, DMIX, and RMIX). We **had conducted** multiple GradPS experiments on **Pymarl3** in previous rebuttal.
> > >
> > > In this rebuttal, we **implement GradPS and others fairly on Pymarl3**. We have run multiple (more than 6) sets of experiments on Pymarl3. The experimental results are shown in the Pymarl3 folder of the [anonymous link](https://anonymous.4open.science/r/review1_1-4871/). GradPS performs better than the competing methods. **These results indicate that GradPS works effectively both for Pymarl and Pymarl3.**
> > >
> > > ## The Necessity of "Futile"
> > >
> > > >Theoretical and experimental results are needed to show difference and relation between "dormant" and "futile".
> > >
> > > We would like to point out that the ICML 2023 Oral paper [2] which introduces the concept of dormant neurons in RL, is an empirical research. It does not involve any theoretical proof. We follow their approach, which focuses on empirical discovery.
> > >
> > > **We show that futile neurons are different from dormant neurons both conceptually and empirically.** We had described the conceptual difference among dormant and futile neurons in detailed table in previous rebuttal.
> > >
> > > In this rebuttal, we report the percentage of dormant neurons and futile neurons in agent networks with VDN during the training process in three simple payoff matrix games (described in Sec. 4.3 and Appendix D.2.3). Moreover, we evaluate the MSE of reconstructing the Payoff matrix when using ReDo[2] and GradPS. ReDo and GradPS are methods developed to reduce the percentage of dormant neurons and futile neurons, respectively. The default parameters of ReDo and GradPS are used.
> > >
> > > The results are shown in Matrix Game folder in [anonymous link](https://anonymous.4open.science/r/review1_1-4871/). The results for each matrix game are located in different sub-folders.
> > >
> > > For the $f_2= f_1 \times 0$ and $f_2=f_1 \times 0.5$ games, the Neuron Percentage graphs show that dormant neurons gradually decrease and stabilize to fix values (around 10%), whereas futile neurons rise to 100% after a few thousand environment steps. For the $f_2= f_1\times -1$ game, the percents of the dormant neurons and the futile neurons fluctuate around 30%-40% and 80%-90%, respectively. **These figures show that the percentages for the two types of neurons are different across training processes.**
> > >
> > > For all three matrix games, the MSE of reconstructed payoff matrix is shown in MSE.pdf figures. For these graphs, the MSE for the original VDN, VDN with ReDo, VDN with GradPS are shown.
> > >
> > > When using ReDo with VDN, the MSE does not change significantly compared to the VDN method. However, when using GradPS with VDN, the MSE drops significantly. For these matrix games, in terms of reducing MSE, GradPS, as a futile-neuron-method, works better than ReDo[2], a dormant-neuron-based method. **These findings further highlight the significant difference among the two types of neurons.**
> > >
> > > ## Presentation of Grouping Method
> > >
> > > >My original review means the grouping method should not be present in only several purely text sentences with any equations.
> > >
> > > We had presented the grouping method through texts and equations (Sec 5.1 and Appendix C.3.), figure (Figure 6), and algorithm (Algorithm 1). We will improve the presentation of our work.
> > >
> > > ## More baselines
> > > To address the concerns of Reviewer EAb8, we have compared GradPS against AdaPS and MADPS in Pymarl. The results shown in the PS-baseline folder of the [anonymous link](https://anonymous.4open.science/r/review1_1-4871/) indicate that GradPS performs better than them.
> > >
> > >
> > > [1] The StarCraft Multi-Agent Challenge.
> > >
> > > [2] The Dormant Neuron Phenomenon in Deep Reinforcement Learning, ICML 2023 Oral

---

### Decision · Program_Chairs · 2025-05-01

**Decision:**

Accept (poster)

**Comment:**

This paper presents a new method for improving parameter sharing in MARL by improving diversity in the shared network. It does this by cloning and updating 'futile' neurons that have conflicting gradients when updated by different agents.

There are some concerns about the significance and practicality of approach but it appears to be novel and beneficial. This line of work should be interesting to the community.

The paper should be improved to clarify details about the method and experiments. The author response was helpful in this regard, but the paper should be updated with the additional details and additional experiments.